# Knowledge-Adaptation Priors

**Mohammad Emtiyaz Khan**[*]
RIKEN Center for AI Project
Tokyo, Japan
emtiyaz.khan@riken.jp

**Siddharth Swaroop**[*]
University of Cambridge
Cambridge, UK
ss2163@cam.ac.uk

## Abstract

Humans and animals have a natural ability to quickly adapt to their surroundings, but machine-learning models, when subjected to changes, often require a complete retraining from scratch. We present Knowledge-adaptation priors (K-priors) to reduce the cost of retraining by enabling quick and accurate adaptation for a wide-variety of tasks and models. This is made possible by a combination of weight and function-space priors to reconstruct the gradients of the past, which recovers and generalizes many existing, but seemingly-unrelated, adaptation strategies. Training with simple first-order gradient methods can often recover the exact retrained model to an arbitrary accuracy by choosing a sufficiently large memory of the past data. Empirical results show that adaptation with K-priors achieves performance similar to full retraining, but only requires training on a handful of past examples.

## 1 Introduction

Machine-Learning (ML) at production often requires constant model updating which can have huge financial and environmental costs [19, 44]. The production pipeline is continuously evolving, where new data are regularly pooled and labeled and old data become irrelevant. Regular tuning of hyperparameters is required to handle drifts [19], and sometimes even the model class/architecture may need to change. Due to this, the model is frequently retrained, retested, and redeployed, which can be extremely costly, especially when the data and model sizes are large. The cost can be reduced if, instead of repeated retraining, the system can quickly adapt to incremental changes. Humans and animals can naturally use their prior knowledge to handle a wide-variety of changes in their surroundings, but such quick, wide, and accurate adaptation has been difficult to achieve in ML.

In theory, this should be possible within a Bayesian framework where the posterior is used as the prior for the future, but exact Bayes is computationally challenging and the design of generic Bayesian priors has its own challenges [55, 39]. In ML, simpler mechanisms are more popular, for example, in Support Vector Machines (SVMs) for adding/removing data [12, 58, 60], and in deep learning for model compression [24]. Weight-priors are used in online learning [13], and more recently for continual learning [30, 40, 47, 33, 62], but they are not suited for many other tasks, such as model compression. In some settings, they also perform worse, for example, in continual learning when compared to memory-based strategies [34, 46, 45, 9]. All these previous works apply to narrow, specific settings, and designing generic adaptation-mechanisms remains an open challenge.

We present Knowledge-adaptation priors (K-priors) for the design of generic adaptation-mechanisms. The general principle of adaptation is to combine the weight and function-space divergences to faithfully reconstruct the gradient of the past. K-priors can handle a wide variety of adaptation tasks (Fig. 1, left) and work for a range of models, such as generalized linear models, deep networks, and their Bayesian extensions. The principle unifies and generalizes many seemingly-unrelated existing works, for example, weight-priors [13], knowledge distillation [24], SVMs [12, 35, 58],

---

[*] Authors contributed equally.

35th Conference on Neural Information Processing Systems (NeurIPS 2021).

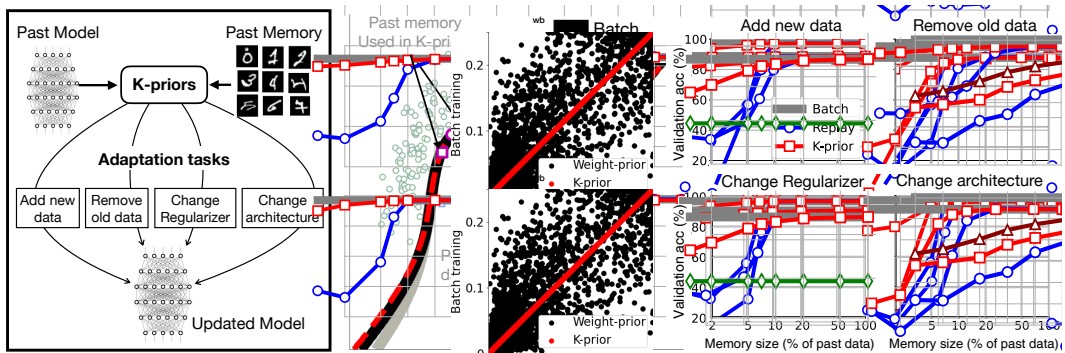

Figure 1: **Left:** K-priors can handle a wide-variety of adaptation tasks by using the past model and a small memory of the past data. **Middle:** For adaptation, the model needs tweaking at only a handful of past examples (shown with dark purple markers). The adapted model (dashed red line) is very close to the retraining on the full batch (solid black line). **Right:** Results on binary classification on 'USPS' digits with neural networks show that K-priors (red square) obtain solutions close to the batch training (gray line) but by using only a fraction (2-5%) of the past data. It also performs better than 'Replay' (blue circles) where the same memory is used for replay. See Sec. 5 for more details.

online Gaussian Processes [17], and continual learning [36, 45, 9]. It leads to natural adaptation-mechanisms where models' predictions need to be readjusted only at a handful of past experiences (Fig. 1, middle). This is quick and easy to train with first-order optimization methods and, by choosing a sufficiently large memory, we obtain results similar to retraining-from-scratch (Fig. 1, right). Code is available at `https://github.com/team-approx-bayes/kpriors`.

## 2 Adaptation in Machine Learning

### 2.1 Knowledge-adaptation tasks

Our goal is to quickly and accurately adapt an already trained model to incremental changes in its training framework. Throughout we refer to the trained model as the *base model*. We denote its outputs by $f_{\mathbf{w}_*}(\mathbf{x})$ for inputs $\mathbf{x} \in \mathbb{R}^D$, and assume that its parameters $\mathbf{w}_* \in \mathcal{W} \subset \mathbb{R}^P$ are obtained by solving the following problem on the data $\mathcal{D}$,

$$\mathbf{w}_* = \underset{\mathbf{w}\in\mathcal{W}}{\arg\min}\ \bar{\ell}(\mathbf{w}), \quad \text{where } \bar{\ell}(\mathbf{w}) = \sum_{i\in\mathcal{D}} \ell_i(\mathbf{w}) + \mathcal{R}(\mathbf{w}). \tag{1}$$

Here, $\ell_i(\mathbf{w})$ denotes the loss function on the $i$'th data example, and $\mathcal{R}(\mathbf{w})$ is a regularizer.

A good adaptation method should be able to handle many types of changes. The simplest and most common change is to add/remove data examples, as shown below at the left where the data example $j \notin \mathcal{D}$ is added to get $\mathbf{w}_+$, and at the right where an example $k \in \mathcal{D}$ is removed to get $\mathbf{w}_-$,

$$\mathbf{w}_+ = \underset{\mathbf{w}\in\mathcal{W}}{\arg\min} \sum_{i\in\mathcal{D}\cup j} \ell_i(\mathbf{w}) + \mathcal{R}(\mathbf{w}), \qquad \mathbf{w}_- = \underset{\mathbf{w}\in\mathcal{W}}{\arg\min} \sum_{i\in\mathcal{D}\setminus k} \ell_i(\mathbf{w}) + \mathcal{R}(\mathbf{w}). \tag{2}$$

We refer to these problems as 'Add/Remove Data' tasks.

Other changes are the 'Change Regularizer' (shown on the left) task where the regularizer is replaced by a new one $\mathcal{G}(\mathbf{w})$, and the 'Change Model Class/Architecture' task (shown on the right) where the model class/architecture is changed leading to a change of the parameter space from $\mathcal{W}$ to $\Theta$,

$$\mathbf{w}_\mathcal{G} = \underset{\mathbf{w}\in\mathcal{W}}{\arg\min} \sum_{i\in\mathcal{D}} \ell_i(\mathbf{w}) + \mathcal{G}(\mathbf{w}), \qquad \boldsymbol{\theta}_* = \underset{\boldsymbol{\theta}\in\Theta}{\arg\min} \sum_{i\in\mathcal{D}} \tilde{\ell}_i(\boldsymbol{\theta}) + \tilde{\mathcal{R}}(\boldsymbol{\theta}). \tag{3}$$

Here, we assume the loss $\tilde{\ell}_i(\boldsymbol{\theta})$ has the same form as $\ell_i(\mathbf{w})$ but using the new model $\tilde{f}_{\boldsymbol{\theta}}(\mathbf{x})$ for prediction, with $\boldsymbol{\theta} \in \Theta$ as the new parameter. $\boldsymbol{\theta}$ can be of a different dimension, and the new regularizer can be chosen accordingly. The change in the model class could be a simple change in

features, for example it is common in linear models to add/remove features, or the change could be similar to model compression or knowledge distillation [24], which may not have a regularizer.

Another type of change is to add *privileged* information, originally proposed by Vapnik and Izmailov [60]. The goal is to include different types of data to improve the performance of the model. This is combined with knowledge distillation by Lopez-Paz et al. [37] and has recently been applied to domain adaptation in deep learning [4, 50, 51].

There could be several other kinds of changes in the training framework, for example, those involving a change in the loss function, or even a combination of the changes discussed above. Our goal is to develop a method that can handle such wide-variety of changes, or 'adaptation tasks' as we will call them throughout. Such adaptation can be useful to reduce the cost of model updating, for example in a continuously evolving ML pipeline. Consider $k$-fold cross-validation, where the model is retrained from scratch for every data-fold and hyperparameter setting. Such retraining can be made cheaper and faster by reusing the model trained in the previous folds and adapting them for new folds and hyperparameters. Model reuse can also be useful during active-learning for dataset curation, where a decision to include a new example can be made by using a quick Add Data adaptation. In summary, model adaptation can reduce the cost by avoiding the need to constantly retrain.

## 2.2 Challenges of knowledge adaptation

Knowledge adaptation in ML has proven to be challenging. Currently there are no methods that can handle many types of adaptation tasks. Most existing works apply narrowly to specific models and mainly focuses on adaptation to Add/Remove Data only. This includes many early proposals for SVMs [12, 59, 25, 20, 49, 35, 31, 58], recent ones for machine-unlearning [11, 21, 41, 22, 53, 8], and methods that use weight and functional priors, for example, for online learning [13], continual deep learning [30, 40, 47, 33, 62, 34, 46, 7, 57, 45, 9], and Gaussian-Process (GP) models [17, 52, 56]. The methodologies of these methods are entirely different from each other, and they do not appear to have any common principles behind their adaptation mechanisms. Our goal is to fill this gap and propose a single method that can handle a wide-variety of tasks for a range of models.

We also note that there is no prior work on the Change Regularizer task. Adaptation has been used to boost hyperparameter tuning [61] but only Add/Remove Data adaptation is used. Warm-starts have been employed as well [18], but it is often not sufficient and can even hurt performance [5].

## 2.3 Problem setting and notation

Throughout, we will use a supervised problem where the loss is specified by an exponential-family,

$$\ell(y, h(f)) = -\log p(y|f) = -\langle y, f \rangle + A(f), \tag{4}$$

where $y \in \mathcal{Y}$ denotes the scalar observation output, $f \in \mathcal{F}$ is the canonical natural parameter, $A(f)$ is the log-partition function, and $h(f) = \mathbb{E}(y) = \nabla A(f)$ is the expectation parameter. A typical example is the cross-entropy loss for binary outcomes $y \in \{0, 1\}$ where $A(f) = \log(1 + e^f)$ and $h(f) = \sigma(f)$ is the Sigmoid function. It is straightforward to extend our method to a vector observation and model outputs. An extension to other types of learning frameworks is discussed in the next section (see the discussion around Eq. 14).

Throughout the paper, we will use a shorthand for the model outputs, where we denote $f_{\mathbf{w}}^i = f_{\mathbf{w}}(\mathbf{x}_i)$. We will repeatedly make use of the following expression for the derivative of the loss,

$$\nabla \ell(y_i, h(f_{\mathbf{w}}^i)) = \nabla f_{\mathbf{w}}^i \, [h(f_{\mathbf{w}}^i) - y_i]. \tag{5}$$

# 3 Knowledge-Adaptation Priors (K-priors)

We present Knowledge-adaptation priors (K-priors) to quickly and accurately adapt a model's knowledge to a wide variety of changes in its training framework. K-priors, denoted below by $\mathcal{K}(\mathbf{w}; \mathbf{w}_*, \mathcal{M})$, refer to a class of priors that use both weight and function-space regularizers,

$$\mathcal{K}(\mathbf{w}; \mathbf{w}_*, \mathcal{M}) = \mathbb{D}_f \left( \mathbf{f}(\mathbf{w}) \| \mathbf{f}(\mathbf{w}_*) \right) + \tau \mathbb{D}_w \left( \mathbf{w} \| \mathbf{w}_* \right), \tag{6}$$

where $\mathbf{f}(\mathbf{w})$ is a vector of $f_{\mathbf{w}}(\mathbf{u}_i)$, defined at inputs in $\mathcal{M} = (\mathbf{u}_1, \mathbf{u}_2, \ldots, \mathbf{u}_M)$. The divergence $\mathbb{D}_f(\cdot \| \cdot)$ measures the discrepancies in the function space $\mathcal{F}$, while $\mathbb{D}_w(\cdot \| \cdot)$ measures the same in the

weight space $\mathcal{W}$. Throughout, we will use Bregman divergences $\mathcal{B}_\psi(p_1, p_2) = \psi(p_1) - \psi(p_2) - \nabla\psi(p_2)^\top(p_1 - p_2)$, specified using a strictly-convex Bregman function $\psi(\cdot)$.

K-priors are defined using the base model $\mathbf{w}_*$, the memory set $\mathcal{M}$, and a trade-off parameter $\tau > 0$. We keep $\tau = 1$ unless otherwise specified. It might also use other parameters required to define the divergence functions. We will sometimes omit the dependency on parameters and refer to $\mathcal{K}(\mathbf{w})$.

Our general principle of adaptation is to use $\mathcal{K}(\mathbf{w})$ to faithfully reconstruct the gradients of the past training objective. This is possible due to the combination of weight and function-space divergences. Below, we illustrate this point for supervised learning for Generalized Linear Models (GLMs).

### 3.1 K-priors for GLMs

GLMs include models such as logistic and Poisson regression, and have a linear model $f_{\mathbf{w}}^i = \phi_i^\top \mathbf{w}$, with feature vectors $\phi_i = \phi(\mathbf{x}_i)$. The base model is obtained as follows,

$$\mathbf{w}_* = \arg\min_{\mathbf{w} \in \mathcal{W}} \sum_{i \in \mathcal{D}} \ell(y_i, h(f_{\mathbf{w}}^i)) + \mathcal{R}(\mathbf{w}). \tag{7}$$

In what follows, for simplicity, we use an $L_2$ regularizer $\mathcal{R}(\mathbf{w}) = \frac{1}{2}\delta\|\mathbf{w}\|^2$, with $\delta > 0$.

We will now discuss a K-prior that *exactly* recovers the gradients of this objective. For this, we choose $\mathbb{D}_w(\cdot\|\cdot)$ to be the Bregman divergence with $\mathcal{R}(\mathbf{w})$ as the Bregman function,

$$\mathbb{D}_w(\mathbf{w}\|\mathbf{w}_*) = \mathcal{B}_\mathcal{R}(\mathbf{w}\|\mathbf{w}_*) = \frac{1}{2}\delta\|\mathbf{w} - \mathbf{w}_*\|^2.$$

We set memory $\mathcal{M} = \mathcal{X}$, where $\mathcal{X}$ is the set of all inputs from $\mathcal{D}$. We regularize each example using separate divergences whose Bregman function is equal to the log-partition $A(f)$ (defined in Eq. 4),

$$\mathbb{D}_f(\mathbf{f}(\mathbf{w})\|\mathbf{f}(\mathbf{w}_*)) = \sum_{i \in \mathcal{X}} \mathcal{B}_A(f_{\mathbf{w}}^i\|f_{\mathbf{w}_*}^i) = \sum_{i \in \mathcal{X}} \ell\left(h(f_{\mathbf{w}_*}^i), h(f_{\mathbf{w}}^i)\right) + \text{constant}.$$

Smaller memories are discussed later in this section. Setting $\tau = 1$, we get the following K-prior,

$$\mathcal{K}(\mathbf{w}; \mathbf{w}_*, \mathcal{X}) = \sum_{i \in \mathcal{X}} \ell\left(h(f_{\mathbf{w}_*}^i), h(f_{\mathbf{w}}^i)\right) + \frac{1}{2}\delta\|\mathbf{w} - \mathbf{w}_*\|^2, \tag{8}$$

which has a similar form to Eq. 7, but the outputs $y_i$ are now replaced by the predictions $h(f_{\mathbf{w}_*}^i)$, and the base model $\mathbf{w}_*$ serves as the mean of a Gaussian weight prior.

We can now show that the gradient of the above K-prior is equal to that of the objective used in Eq. 7,

$$\nabla\mathcal{K}(\mathbf{w}; \mathbf{w}_*, \mathcal{X}) = \sum_{i \in \mathcal{X}} \phi_i\left[h(f_{\mathbf{w}}^i) - h(f_{\mathbf{w}_*}^i)\right] + \delta(\mathbf{w} - \mathbf{w}_*), \tag{9}$$

$$= \underbrace{\sum_{i \in \mathcal{D}} \phi_i\left[h(f_{\mathbf{w}}^i) - y_i\right] + \delta\mathbf{w}}_{=\nabla\ell(\mathbf{w}).} - \underbrace{\sum_{i \in \mathcal{D}} \phi_i\left[h(f_{\mathbf{w}_*}^i) - y_i\right] - \delta\mathbf{w}_*}_{=0.}, \tag{10}$$

where the first line is obtained by using Eq. 5 and noting that $\nabla f_{\mathbf{w}}^i = \phi_i$, and the second line is obtained by adding and subtracting outputs $y_i$ in the first term. The second term there is equal to 0 because $\mathbf{w}_*$ is a minimizer and therefore $\nabla\ell(\mathbf{w}_*) = 0$. In this case, the K-prior with $\mathcal{M} = \mathcal{X}$ exactly recovers the gradient of the past training objective.

Why are we able to recover exact gradients? This is because the structure of the K-prior closely follows the structure of Eq. 7: the gradient of each term in Eq. 7 is recovered by a corresponding divergence in the K-prior. The gradient recovery is due to the property that the gradient of a Bregman divergence is the difference between the *dual* parameters $\nabla\psi(p)$:

$$\nabla_{p_1}\mathcal{B}(p_1, p_2) = \nabla\psi(p_1) - \nabla\psi(p_2).$$

This leads to Eq. 9. For the function-space divergence term, $h(f_{\mathbf{w}}^i) - h(f_{\mathbf{w}_*}^i)$ are the differences in the (dual) expectation parameters. For the weight-space divergence term, we note that the dual space is equal to the original parameter space $\mathcal{W}$ for the $L_2$ regularizer, leading to $\mathbf{w} - \mathbf{w}_*$. Lastly, we find that terms cancel out by using the optimality of $\mathbf{w}_*$, giving us the exact gradients.

## 3.2 K-priors with limited memory

In practice, setting $\mathcal{M} = \mathcal{X}$ might be as slow as full retraining, but for incremental changes, we may not need all of them (see Fig. 1, for example). Then, which inputs should we include? The answer lies in the gradient-reconstruction error $\mathbf{e}$, shown below for $\mathcal{M} \subset \mathcal{X}$,

$$\mathbf{e}(\mathbf{w}; \mathcal{M}) = \nabla \bar{\ell}(\mathbf{w}) - \nabla \mathcal{K}(\mathbf{w}; \mathbf{w}_*, \mathcal{M}) = \sum_{i \in \mathcal{X} \backslash \mathcal{M}} \boldsymbol{\phi}_i \left[ h(f_{\mathbf{w}}^i) - h(f_{\mathbf{w}_*}^i) \right]. \tag{11}$$

The error depends on the "leftover" $\boldsymbol{\phi}_i$ for $i \in \mathcal{X} \backslash \mathcal{M}$, and their discrepancies $h(f_{\mathbf{w}}^i) - h(f_{\mathbf{w}_*}^i)$. A simple idea could be to include the inputs where predictions disagree the most, but this is not feasible because the candidates $\mathbf{w}$ are not known beforehand. The following approximation is more practical,

$$\mathbf{e}(\mathbf{w}; \mathcal{M}) \approx \mathbf{G}_*(\mathcal{X} \backslash \mathcal{M})(\mathbf{w} - \mathbf{w}_*), \text{ where } \mathbf{G}_*(\mathcal{X} \backslash \mathcal{M}) = \sum_{i \in \mathcal{X} \backslash \mathcal{M}} \boldsymbol{\phi}_i h'(f_{\mathbf{w}_*}^i) \boldsymbol{\phi}_i^\top. \tag{12}$$

This is obtained by using the Taylor approximation $h(f_{\mathbf{w}}^i) - h(f_{\mathbf{w}_*}^i) \approx h'(f_{\mathbf{w}_*}^i)(\nabla f_{\mathbf{w}_*}^i)^\top (\mathbf{w} - \mathbf{w}_*)$ in Eq. 11 ($h'(f^i)$ is the derivative). The approximation is conveniently expressed in terms of the Generalized Gauss-Newton (GGN) matrix [38], denoted by $\mathbf{G}_*(\cdot)$. The approximation suggests that $\mathcal{M}$ should be chosen to keep the *leftover* GGN matrix $\mathbf{G}_*(\mathcal{X} \backslash \mathcal{M})$ orthogonal to $\mathbf{w} - \mathbf{w}_*$. Since $\mathbf{w}$ changes during training, a reasonable approximation is to choose examples that keep the top-eigenvalues of the GGN matrix. This can be done by forming a low-rank approximation by using sketching methods, such as the leverage score [16, 1, 15, 10]. A cheaper alternative is to choose the examples with highest $h'(f_{\mathbf{w}_*}^i)$. The quantity is cheap to compute in general, for example, for deep networks it is obtained with just a forward pass. Such a set has been referred to as the 'memorable past' by Pan et al. [45], who found it to work well for classification. Due to its simplicity, we will use this method in our experiments, and leave the application of sketching methods as future work.

K-priors with limited memory can achieve low reconstruction error. This is due to an important feature of K-priors: they do not restrict inputs $\mathbf{u}_i$ to lie within the training set $\mathcal{X}$. The inputs can be arbitrary locations in the input space. This still works because a ground-truth label is not needed for $\mathbf{u}_i$, and only model predictions $f_{\mathbf{w}}(\mathbf{u}_i)$ are used in K-priors. As long as the chosen $\mathbf{u}_i$ represent $\mathcal{X}$ well, we can achieve a low gradient-reconstruction error, and sometimes even perfect reconstruction. Theoretical results regarding this point are discussed in App. A, where we present the *optimal* K-prior which can theoretically achieve perfect reconstruction error by using singular vectors of $\mathbf{\Phi}^\top = [\boldsymbol{\phi}(\mathbf{x}_1), \boldsymbol{\phi}(\mathbf{x}_2), \ldots, \boldsymbol{\phi}(\mathbf{x}_N)]$. When only top-$M$ singular vectors are chosen, the error grows according to the leftover singular values. The optimal K-prior is difficult to realize in practice, but the result shows that it is theoretically possible to achieve low error with limited memory.

## 3.3 Adaptation using K-priors

We now discuss how the K-prior of Eq. 8 can be used for the Add/Remove Data tasks. Other adaptation tasks and corresponding K-priors are discussed in App. B.

Because K-priors can reconstruct the gradient of $\bar{\ell}(\mathbf{w})$, we can use them to adapt instead of retraining from scratch. For example, to add/remove data from the GLM solution in Eq. 7, we can use the following K-prior regularized objectives,

$$\hat{\mathbf{w}}_+ = \underset{\mathbf{w} \in \mathcal{W}}{\arg\min} \ \ell_j(\mathbf{w}) + \mathcal{K}(\mathbf{w}; \mathbf{w}_*, \mathcal{M}), \qquad \hat{\mathbf{w}}_- = \underset{\mathbf{w} \in \mathcal{W}}{\arg\min} \ -\ell_k(\mathbf{w}) + \mathcal{K}(\mathbf{w}; \mathbf{w}_*, \mathcal{M}). \tag{13}$$

Using Eq. 10, it is easy to show that this recovers the exact solution when all the past data is used. App. B details analogous results for the Change Regularizer and Change Model Class tasks.

**Theorem 1.** *For $\mathcal{M} = \mathcal{X}$, we have $\mathbf{w}_+ = \hat{\mathbf{w}}_+$ and $\mathbf{w}_- = \hat{\mathbf{w}}_-$.*

For limited memory, we expect the solutions to be close when memory is large enough. This is because the error in the gradient is given by Eq. 11. The error can be reduced by choosing better $\mathcal{M}$ and/or by increasing its size to ultimately get perfect recovery.

We stress that Eq. 13 is fundamentally different from replay methods that optimize an objective similar to Eq. 2 but use a small memory of past examples [48]. Unlike such methods, we use the predictions $h(f_{\mathbf{w}_*}^i)$, which we can think of as soft labels, with potentially more information than the true one-hot encoded labels $y_i$. Given a fixed memory budget, we expect K-prior regularization to perform better than such replay methods, and we observe this empirically in Sec. 5.

## 3.4 K-priors for Generic Learning Problems

The main principle behind the design of K-priors is to construct it such that the gradients can faithfully be reconstructed. As discussed earlier, this is often possible by exploiting the structure of the learning problem. For example, to replace an old objective such as Eq. 1, with loss $\ell_i^{\mathrm{old}}(f)$ and regularizer $\mathcal{R}^{\mathrm{old}}(\mathbf{w})$, with a new objective with loss $\ell_i^{\mathrm{new}}(f)$ and regularizer $\mathcal{R}^{\mathrm{new}}(\mathbf{w})$, the divergences should be chosen such that they have the following gradients,

$$\nabla\mathbb{D}_w(\mathbf{w}\|\mathbf{w}_*) = \nabla\mathcal{R}^{\mathrm{new}}(\mathbf{w}) - \nabla\mathcal{R}^{\mathrm{old}}(\mathbf{w}), \qquad \nabla\mathbb{D}_f(\mathbf{f}(\mathbf{w})\|\mathbf{f}(\mathbf{w}_*)) = \nabla\mathbf{f}(\mathbf{w})^{\top}\mathbf{B}\,\mathbf{d}_m \qquad (14)$$

where $\mathbf{d}_m$ is an $M$-length vector with the discrepancy $\nabla\ell_i^{\mathrm{new}}(f_{\mathbf{w}}^i) - \nabla\ell_i^{\mathrm{old}}(f_{\mathbf{w}_*}^i)$ as the $i$'th entry. The matrix $\mathbf{B}$ is added to counter the mismatch between $\mathcal{D}$ and $\mathcal{M}$. Similar constructions can be used for other learning objectives. For non-differentiable functions, a Bayesian version can be used with the Kullback-Leibler (KL) divergence (we discuss an example in the next section). We can use exponential-family distributions which implies a Bregman divergence through KL [6]. Since the gradient of such divergences is equal to the difference in the dual-parameters, the general principle is to use divergences with an appropriate dual space to swap the old information with new information.

## 4 K-priors: Extensions and Connections

The general principle of adaptation used in K-priors connects many existing works in specific settings. We will now discuss these connections to show that K-priors provide a unifying and generalizing principle for these seemingly unrelated methods in fields such as online learning, deep learning, SVMs, Bayesian Learning, Gaussian Processes, and continual learning.

### 4.1 Weight-Priors

Quadratic or Gaussian weight-priors [13, 30, 54] be seen as as specialized cases of K-priors, where restrictive approximations are used. For example, the following quadratic regularizer,

$$\mathcal{R}_{\mathrm{quad}}(\mathbf{w};\mathbf{w}_*) = (\mathbf{w} - \mathbf{w}_*)^{\top}\left[\mathbf{G}_*(\mathcal{X}) + \delta\mathbf{I}\right](\mathbf{w} - \mathbf{w}_*),$$

which is often used in used in online and continual learning [13, 30], can be seen as a first-order approximation of the K-prior in Eq. 8. This follows by approximating the K-prior gradient in Eq. 8 by using the Taylor approximation used in Eq. 12, to get

$$\nabla\mathcal{K}(\mathbf{w};\mathbf{w}_*,\mathcal{X}) \approx \sum_{i\in\mathcal{X}}\boldsymbol{\phi}_i\left[h'(\boldsymbol{\phi}_i^{\top}\mathbf{w}_*)\boldsymbol{\phi}_i^{\top}(\mathbf{w} - \mathbf{w}_*)\right] + \delta(\mathbf{w} - \mathbf{w}_*)\ = \nabla\mathcal{R}_{\mathrm{quad}}(\mathbf{w};\mathbf{w}_*).$$

K-priors can be more accurate than weight priors but may require larger storage for the memory points. However, we expect the memory requirements to grow according to the rank of the feature matrix (see App. A) which may still be manageable. If not, we can apply sketching methods.

### 4.2 K-priors for Deep Learning and Connections to Knowledge Distillation

We now discuss the application to deep learning. It is clear that the functional term in K-priors is similar to Knowledge distillation (KD) [24], which is a popular approach for model compression in classification problems using a softmax function (the following expression is from Lopez-Paz et al. [37], also see App. E),

$$\ell_{\mathrm{KD}}(\mathbf{w}) = \lambda\sum_{i\in\mathcal{D}}\ell\left(y_i, h(f_{\mathbf{w}}^i)\right) + (1-\lambda)\sum_{i\in\mathcal{D}}\ell\left(h(f_{\mathbf{w}_*}^i/T),\, h(f_{\mathbf{w}}^i)\right), \qquad (15)$$

The base model predictions are often scaled with a temperature parameter $T > 0$, and $\lambda \in [0, 1]$. KD can be seen as a special case of K-priors without the weight-space term ($\tau = 0$). K-priors extend KD in three ways, by (i) adding the weight-space term, (ii) allowing general link functions or divergence functions, and (iii) using a potentially small number of examples in $\mathcal{M}$ instead of the whole dataset. With these extensions, K-priors can handle adaptation tasks other than compression. Due to their similarity, it is also possible to borrow tricks used in KD to improve the performance of K-priors.

KD often yields solutions that are better than retraining from scratch. Theoretically the reasons behind this are not understood well, but we can view KD as a mechanism to reconstruct the past

gradients, similarly to K-priors. As we now show, this gives a possible explanation behind KD's success. Unlike GLMs, K-priors for deep learning do not recover the exact gradient of the past training objective, and there is an additional left-over term (a derivation is in App. C),

$$\nabla \mathcal{K}(\mathbf{w}) = \underbrace{\sum_{i \in \mathcal{D}} \nabla f_{\mathbf{w}}^i \left[ h(f_{\mathbf{w}}^i) - y_i \right] + \delta \mathbf{w}}_{= \nabla \bar{\ell}(\mathbf{w})} - \underbrace{\left( \sum_{i \in \mathcal{D}} \nabla f_{\mathbf{w}}^i r_{\mathbf{w}_*}^i + \delta \mathbf{w}_* \right)}_{\text{Additional term since } \nabla f_{\mathbf{w}}^i \neq \nabla f_{\mathbf{w}_*}^i}, \tag{16}$$

where $r_{\mathbf{w}_*}^i := h(f_{\mathbf{w}_*}^i) - y_i$ is the residual of the base model. It turns out that the gradient of the KD objective in Eq. 15 has this exact same form when $\delta = 0, T = 1$ (derivation in App. C),

$$\nabla \ell_{\text{KD}}(\mathbf{w}) = \sum_{i \in \mathcal{D}} \nabla f_{\mathbf{w}}^i \left[ h(f_{\mathbf{w}}^i) - y_i \right] - (1 - \lambda) \sum_{i \in \mathcal{D}} \nabla f_{\mathbf{w}}^i r_{\mathbf{w}_*}^i.$$

The additional term adds large gradients to push away from the high residual examples (the examples the teacher did not fit well). This is similar to Similarity-Control for SVMs from Vapnik and Izmailov [60], where "slack"-variables are used in a dual formulation to improve the student, who could now be solving a simpler separable classification problem. The residuals $r_{\mathbf{w}_*}^i$ above play a similar role as the slack variables, but they do not require a dual formulation. Instead, they arise due to the K-prior regularization in a primal formulation. In this sense, K-priors can be seen as an easy-to-implement scheme for Similarity Control, that could potentially be useful for student-teacher learning.

Lopez-Paz et al. [37] use this idea further to generalize distillation and interpret residuals from the teachers as corrections for the student (see Eq. 6 in their paper). In general, it is desirable to trust the knowledge of the base model and use it to improve the adapted model. These previous ideas are now unified in K-priors: we can provide the information about the decision boundary to the student in a more accessible form than the original data (with true labels) could.

### 4.3 Adding/removing data for SVMs

K-prior regularized training yields equivalent solutions to the adaptation strategies used in SVM to add/remove data examples. K-priors can be shown to be equivalent to the primal formulation of such strategies [35]. The key trick to show the equivalence is to use the representer theorem which we will now illustrate for the 'Add Data' task in Eq. 13. Let $\boldsymbol{\Phi}_+$ be the $(N + 1) \times P$ feature matrix obtained on the dataset $\mathcal{D} \cup j$, then by the representer theorem we know that there exists a $\boldsymbol{\beta} \in \mathbb{R}^{N+1}$ such that $\mathbf{w}_+ = \boldsymbol{\Phi}_+^\top \boldsymbol{\beta}$. Taking the gradient of Eq. 13, and multiplying by $\boldsymbol{\Phi}_+$, we can write the optimality condition as,

$$0 = \boldsymbol{\Phi}_+^\top \nabla [\ell_j(\mathbf{w}_+) + \mathcal{K}(\mathbf{w}_+)] = \sum_{i \in \mathcal{D} \cup j} \left( \nabla_f \ell(y_i, h(f))|_{f = \boldsymbol{\beta}_i^\top \mathbf{k}_{i,+}} \right) \mathbf{k}_{i,+} + \delta \mathbf{K}_+ \boldsymbol{\beta}, \tag{17}$$

where $\mathbf{K}_+ = \boldsymbol{\Phi}_+ \boldsymbol{\Phi}_+^\top$ and its $i$'th column is denoted by $\mathbf{k}_{i,+}$. This is exactly the gradient of the primal objective in the function-space defined over the full batch $\mathcal{D} \cup j$; see Equation 3.6 in Chapelle [14]. The primal strategy is equivalent to the more common dual formulations [12, 59, 25, 20, 49, 31, 58]. The function-space formulations could be computationally expensive, but speed-ups can be obtained by using support vectors. This is similar to the idea of using limited memory in K-priors in Sec. 3.

### 4.4 K-priors for Bayesian Learning and Connections to GPs

K-priors can be seamlessly used for adaptation within a Bayesian learning framework. Consider a Gaussian approximation $q_*(\mathbf{w})$ trained on a variational counterpart of Eq. 1 with prior $p(\mathbf{w}) \propto \exp[-\mathcal{R}(\mathbf{w})]$, and its adapted version where we add data, as shown below ($\mathbb{D}_{\text{KL}}[\cdot \| \cdot]$ is the KL divergence),

$$q_*(\mathbf{w}) = \arg\min_{q \in Q} \sum_{i \in \mathcal{D}} \mathbb{E}_q[\bar{\ell}_i(\mathbf{w})] + \mathbb{D}_{\text{KL}}[p \| q], \quad q_+(\mathbf{w}) = \arg\min_{q \in Q} \sum_{i \in \mathcal{D} \cup j} \mathbb{E}_q[\ell_i(\mathbf{w})] + \mathbb{D}_{\text{KL}}[p \| q].$$

Assuming the same setup as Sec. 3, we can recover $q_+(\mathbf{w})$ by using $q_{\mathcal{K}}(\mathbf{w}) \propto \exp[-\mathcal{K}(\mathbf{w})]$ where we use the K-prior defined in Eq. 8 (note that normalization constant of $q_{\mathcal{K}}$ is not required),

$$\hat{q}_+(\mathbf{w}) = \arg\min_{q \in Q} \mathbb{E}_q[\ell_j(\mathbf{w})] + \mathbb{D}_{\text{KL}}[q \| q_{\mathcal{K}}], \tag{18}$$

This follows using Eq. 10. Details are in App. D. In fact, when this Bayesian extension is written in the function-space similarly to Eq. 17, it is related to the online updates used in GPs [17]. When $q_{\mathcal{K}}$ is built with limited memory, as described in Sec. 3, the application is similar to sparse variational GPs, but now data examples are used as inducing inputs. These connections are discussed in more detail in App. D. Our K-prior formulations operates in the weight-space and can be easily trained with first-order methods, however an equivalent formulation in the function space can also be employed, as is clear from these connections. The above extensions can be extended to handle arbitrary exponential-family approximations by appropriately defining K-priors using KL divergences. We omit these details since this topic is more suitable for a Bayesian version of this paper.

### 4.5 Memory-Based Methods for Deep Continual Learning

K-priors is closely related to recent functional regularization approaches proposed for deep continual learning [34, 46, 7, 54, 57, 45, 9]. The recent FROMP approach of Pan et al. [45] is closest to ours where the form of the functional divergence used is similar to our suggestion in Eq. 14. Specifically, comparing with Eq. 14, their functional divergence correspond to the vector $\mathbf{d}_m$ with the $i$'th entry as $h(f_{\mathbf{w}}(\mathbf{u}_i)) - h(f_{\mathbf{w}_*}(\mathbf{u}_i))$ for $\mathbf{u}_i \in \mathcal{M}$, and the matrix $\mathbf{B}$ is (can be seen as Nystrom approximation),

$$\mathbf{B} = \mathbf{\Lambda}(\mathbf{w}) \left[ \mathbf{\Lambda}(\mathbf{w}_*) \nabla \mathbf{f}(\mathbf{w}_*) \mathbf{G}(\mathbf{w}_*)^{-1} \nabla \mathbf{f}(\mathbf{w}_*)^{\top} \mathbf{\Lambda}(\mathbf{w}_*) \right]^{-1},$$

where $\mathbf{\Lambda}(\mathbf{w})$ is a diagonal matrix with $h'(f_{\mathbf{w}}(\mathbf{u}_i))$ as the $i$'th diagonal entry, and $\mathbf{G}(\mathbf{w}_*) = \nabla \mathbf{f}(\mathbf{w}_*)^{\top} \mathbf{\Lambda}(\mathbf{w}_*) \nabla \mathbf{f}(\mathbf{w}_*) + \delta \mathbf{I}$ is the GGN matrix. They also propose to use 'memorable past' examples obtained by sorting $h'(f_{\mathbf{w}_*}^i)$, which is consistent with our theory (see Eq. 11). Based on our work, we can interpret the approach of Pan et al. [45] as a mechanism to reconstruct the gradient of the past, which gives very good performance in practice.

Another related approach is the gradient episodic memory (GEM) [36], where the goal is to ensure that the $\sum_{i \in \mathcal{M}} [\ell(y_i, f_{\mathbf{w}}^i) - \ell(y_i, f_{\mathbf{w}_*}^i)] < 0$. This is similar in spirit to the student-teacher transfer of Vapnik and Izmailov [60] where the loss of the student is regularized using the model output of the teacher (see Eq. 7 in Vapnik and Izmailov [60] for an example). Lopez-Paz and Ranzato [36] relax the optimization problem to write it in terms of the gradients, which is similar to K-priors, except that K-priors use a first-order optimization method, which is simpler than the dual approach used in Lopez-Paz and Ranzato [36].

Most of these approaches do not employ a weight-space divergence, and sometimes even the function-space divergence is replaced by the Euclidean one [7, 9]. Often, the input locations are sampled randomly, or using a simple replay method [9] which could be suboptimal. Some approaches propose computationally-expensive methods for choosing examples to store in memory [2, 3], and some can be seen as related to choosing points with high leverage [3]. The approach in Titsias et al. [57] uses inducing inputs which is closely connected to the online GP update. The method we proposed does not contradict with these, but gives a more direct way to choose the points where the gradient errors are taken into consideration.

## 5 Experimental Results

We compare the performance of K-priors to retraining with full-batch data ('Batch') and a retraining with replay from a small memory ('Replay'), and use $\tau = 1$. For fair comparisons, we use the same memory for Replay and K-priors obtained by choosing points with highest $h'(f_{\mathbf{w}_*}^i)$ (see Sec. 3). Memory chosen randomly often gives much worse results and we omit these results. Replay uses the true label while K-priors use model-predictions. We compare these three methods on the four adaptation tasks: 'Add Data', 'Remove Data', 'Change Regularizer', and 'Change Architecture'. For the 'Add Data' task, we also compare to Weight-Priors with GGN.

Our overall finding is that, for GLMs and deep learning on small problems, K-priors can achieve the same performance as Batch, but with a small fraction of data (often 2-10% of the full data (Fig. 1, right, and Fig. 2). Replay does much worse for small memory size, which clearly shows the advantage of using the model predictions (instead of true labels) in K-priors. Weight priors generally perform well, but they can do badly when the adaptation involves a drastic change for examples in $\mathcal{M}$ (see Fig. 3). Finally, for large deep-learning problems, results are promising but more investigations are required with extensive hyperparameter tuning.

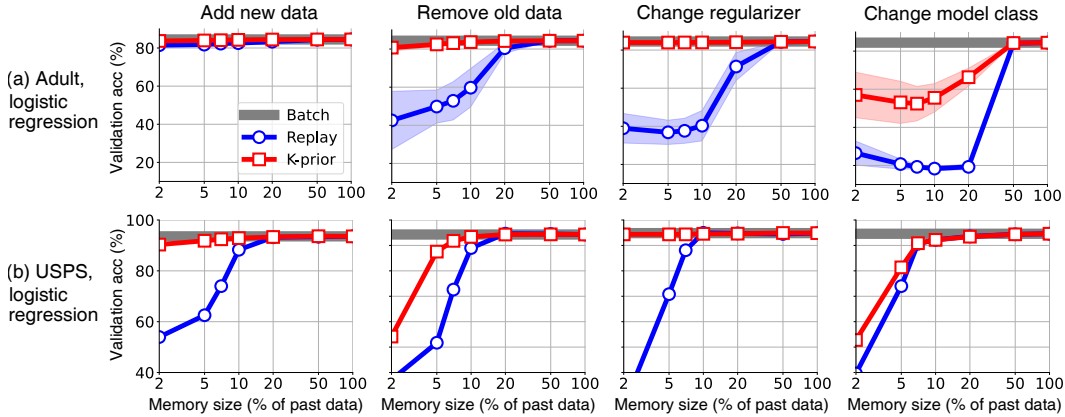

Figure 2: K-priors (red squares) match Batch (grey) while mostly using 2-5% of the data (for only 3 tasks a larger fraction is required). K-priors always outperforms Replay which uses the true labels. K-priors replace the labels by the model predictions (see the discussion after Theorem 1).

Several additional experiments are in App. E. In App. E.4, we study the effect of randomly initialization and find that it performs similarly to a warm start at $\mathbf{w}_*$. In App. E.5, we find that K-priors with limited memory are take much less time to reach a specified accuracy than both batch and replay. The low cost is due to a small memory size. Replay also uses small memory but performs poorly.

**Logistic Regression on the 'UCI Adult' dataset.** This is a binary classification problem consisting of 16,100 examples to predict income of individuals. We randomly sample 10% of the training data (1610 examples), and report mean and standard deviation over 10 such splits. For training, we use the L-BFGS optimizer for logistic regression with polynomial basis. Results are summarized in Fig. 2(a). For the 'Add Data' task, the base model uses 9% of the data and we add 1% new data. For 'Remove Data', we remove 100 data examples (6% of the training set) picked by sorting $h'(f_{\mathbf{w}_*}^i)$. For the 'Change Regularizer' task, we change the $L_2$ regularizer from $\delta = 50$ to $5$, and for 'Change Model Class', we reduce the polynomial degree from $2$ to $1$. K-priors perform very well on the first three tasks, remaining very close to Batch, even when the memory sizes are down to 2%. 'Changing Model Class' is slightly challenging, but K-priors still significantly out-perform Replay.

**Logistic Regression on the 'USPS odd vs even' dataset.** The USPS dataset consists of 10 classes (one for each digit), and has 7,291 training images of size $16 \times 16$. We split the digits into two classes: odd and even digits. Results are in Fig. 2(b). For the 'Add Data' task, we add all examples for the digit 9 to the rest of the dataset, and for 'Remove Data' we remove the digit 8 from the whole dataset. By adding/removing an entire digit, we enforce an *inhomogeneous* data split, making the tasks more challenging. The 'Change Regularizer' and 'Change Model Class' tasks are the same as the Adult dataset. K-priors perform very well on the 'Add Data' and 'Change Regularizer' tasks, always achieving close to Batch performance. For 'Remove Data', which is a challenging task due to inhomogeneity, K-priors still only need to store 5% of past data to maintain close to 90% accuracy, whereas Replay requires 10% of the past data.

**Neural Networks on the 'USPS odd vs even' dataset.** This is a repeat of the previous experiment but with a neural network (a 1-hidden-layer MLP with 100 units). Results are in Fig. 1 (right). The 'Change Regularizer' task now changes $\delta = 5$ to $10$, and the 'Change Architecture' task compresses the architecture from a 2-hidden-layer MLP (100 units per layer) to a 1-hidden-layer MLP with 100 units. We see that even with neural networks, K-priors perform very well, similarly out-performing Replay and remaining close to the Batch solution at small memory sizes.

**Weight-priors vs K-priors.** As discussed in the main text, weight-priors can be seen as an approximation of K-priors where $h'(f_{\mathbf{w}}^i)$ are replaced by 'stale' $h'(f_{\mathbf{w}_*}^i)$, evaluated at the old $\mathbf{w}_*$. In Fig. 3(a), we visualize these 'stale' $h'(f_{\mathbf{w}_*}^i)$ and compare them to K-priors which obtains values close to the ones found by Batch. Essentially, for the points at the diagonal the match is perfect, and we see that it is the case for K-priors but not for the weight-priors. We use logistic regression on the USPS data (the 'Add Data' task). This inhomogeneous data split is difficult for weight-priors, and we show in Fig. 3(b) that weight-priors do not perform well. For homogeneous data splits, weight-priors do

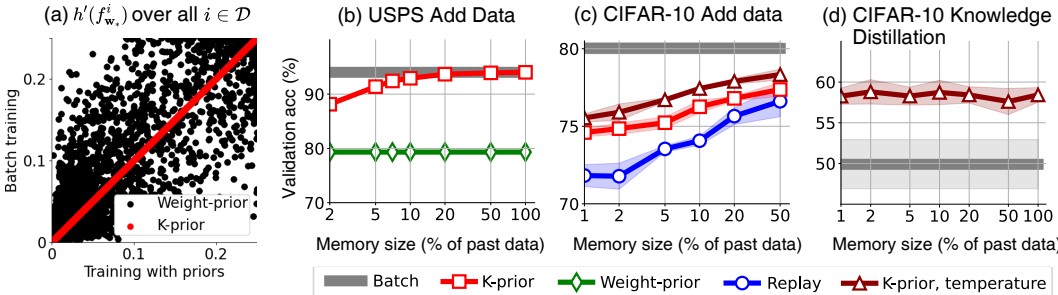

Figure 3: (a) When compared at the Batch solution for the 'Add Data' task on USPS, weight priors give incorrect values of $h'(f_{\mathbf{w}}^i)$ (shown with black dots, each dot corresponds to a data example). Points on the diagonal means a perfect match which is the case for K-priors (show with red dots). (b) Due to this, weight-priors (green diamonds) perform worse than K-priors (red squares). (c) For the 'Add data' task on CIFAR-10, K-priors outperform replay (blue circles), but performance can still be improved by using a temperature parameter (dark-red triangles). (d) The same is true for knowledge distillation [24], and we see that we can reduce memory size while still performing better than the student model.

better than this, but this result goes to show that they do not have any mechanisms to fix the mistakes made in the past. In K-priors, we can always change $\mathcal{M}$ to improve the performance. We provide more plots and results in App. E, including results for weight-priors on all the 'Add data' tasks we considered in this paper.

**MNIST and CIFAR-10, neural networks.** Finally, we discuss results on larger problems in deep learning. We show many adaptation tasks in App. E for 10-way classification on MNIST [32] with MLPs and 10-way classification on CIFAR-10 with CifarNet [62], trained with the Adam optimizer [29]. In Fig. 3(c) we show one representative result for the 'Add data' task with CIFAR-10, where we add a random 10% of CIFAR-10 training data to the other 90% (mean and standard deviation over 3 runs). Although vanilla K-priors outperform Replay, there is now a bigger gap between K-prior and Batch even with 50% past data stored. However, when we use a temperature (similar to knowledge distillation in (15) but with the weight term included), K-priors improves.

A similar result is shown in Fig. 3(d) for knowledge distillation ($\delta = 0$ but with a temperature parameter) where we are distill from a CifarNet teacher to a LeNet5-style student (details in App. E). Here, K-priors with 100% data is equivalent to Knowledge Distillation, but when we reduce the memory size using our method, we still outperform Batch (which is trained from scratch on all data). Overall, our initial effort here suggests that K-priors can do better than Replay, and have potential to give better results with more hyperparameter tuning.

# 6 Discussion

In this paper, we proposed a class of new priors, called K-prior. We show general principles of obtaining accurate adaptation with K-priors which are based on accurate gradient reconstructions. The prior applies to a wide-variety of adaptation tasks for a range of models, and helps us to connect many existing, seemingly-unrelated adaptation strategies in ML. Based on our adaptation principles, we derived practical methods to enable adaptation by tweaking models' predictions at a few past examples. This is analogous to adaptation in humans and animals where past experiences is used for new situations. In practice, the amount of required past memory seems sufficiently low.

The financial and environmental costs of retraining are a huge concern for ML practitioners, which can be reduced with quick adaptations. The current pipelines and designs are specialized for an offline, static setting. Our approach here pushes towards a simpler design which will support a more dynamic setting. The approach can eventually lead to new systems that learn quickly and flexibly, and also act sensibly across a wide range of tasks. This opens a path towards systems that learn incrementally in a continual fashion, with the potential to fundamentally change the way ML is used in scientific and industrial applications. We hope that this work will help others to do more towards this goal in the future. We ourselves will continue to push this work in that direction.

## Acknowledgements

We would like to thank the members of the Approximate-Bayesian-Inference team at RIKEN-AIP. Special thanks to Dr. Thomas Möllenhoff (RIKEN-AIP), Dr. Gian Maria Marconi (RIKEN-AIP), Peter Nickl (RIKEN-AIP), and also to Prof. Richard E. Turner (University of Cambridge). Mohammad Emtiyaz Khan is partially supported by KAKENHI Grant-in-Aid for Scientific Research (B), Research Project Number 20H04247. Siddharth Swaroop is partially supported by a Microsoft Research EMEA PhD Award.

## Author Contributions Statement

List of Authors: Mohammad Emtiyaz Khan (M.E.K.), Siddharth Swaroop (S.S.).

Both the authors were involved in the idea conception. S.S. derived a version of theorem 1 with some help from M.E.K. This was then modified and generalized by M.E.K. for generic adaptation tasks. The general principle of adaptation described in the paper are due to M.E.K., who also derived connections to SVMs and GPs, and extensions to Bayesian settings (with regular feedback from S.S.). Both authors worked together on the connections to Knowledge Distillation and Deep Continual Learning. S.S. performed all the experiments (with regular feedback from M.E.K.). M.E.K. wrote the main sections with the help of S.S., and S.S. wrote the section about the experiments. Both authors proof-read and reviewed the paper.

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
