# A Optimal K-priors for GLMs

We present theoretical results to show that K-priors with limited memory can achieve low gradient-reconstruction error. We will discuss the *optimal* K-prior which can theoretically achieve perfect reconstruction error. Note that the prior is difficult to realize in practice since it requires all past training-data inputs $\mathcal{X}$. Our goal here is to establish a theoretical limit, not to give practical choices.

Our key idea is to choose a few input locations that provide a good representation of the training-data inputs $\mathcal{X}$. We will make use of the singular-value decomposition (SVD) of the feature matrix,

$$\mathbf{\Phi}^\top = \mathbf{U}_{1:K}^* \mathbf{S}_{1:K}^* (\mathbf{V}_{1:K}^*)^\top$$

where $K \leq \min(N, P)$ is the rank, $\mathbf{U}_{1:K}^*$ is $P \times K$ matrix of left-singular vectors $\mathbf{u}_i^*$, $\mathbf{V}_{1:K}^*$ is $N \times K$ matrix of right-singular vectors $\mathbf{v}_i^*$, and $\mathbf{S}_{1:K}^*$ is a diagonal matrix with singular values $s_i$ as the $i$'th diagonal entry.

We define $\mathcal{M}^* = \{\mathbf{u}_1^*, \mathbf{u}_2^*, \ldots, \mathbf{u}_K^*\}$, and the following K-prior,

$$\mathcal{K}_{\text{opt}}(\mathbf{w}; \mathbf{w}_*, \mathcal{M}^*) = \sum_{j=1}^K \beta_j^* \ell\left(h(f_{\mathbf{w}_*}(\mathbf{u}_j^*)), h(f_{\mathbf{w}}(\mathbf{u}_j^*))\right) + \tfrac{1}{2}\delta\|\mathbf{w} - \mathbf{w}_*\|^2. \tag{19}$$

Here, each functional divergence is weighted by $\beta_j^*$ which refers to the elements of the following,

$$\boldsymbol{\beta}^* = \mathbf{D}_u^{-1} \mathbf{S}_{1:K}^* \mathbf{V}_{1:K}^\top \mathbf{d}_x$$

where $\mathbf{d}_x$ is an $N$-length vector with entries $h(f_{\mathbf{w}}^i) - h(f_{\mathbf{w}_*}^i)$ for all $i \in \mathcal{X}$, while $\mathbf{D}_u$ is a $K \times K$ diagonal matrix with diagonal entries $h(f_{\mathbf{w}}(\mathbf{u}_j^*)) - h(f_{\mathbf{w}_*}(\mathbf{u}_j^*))$ for all $j = 1, 2, \ldots, K$. The above definition departs slightly from the original definition where only a single $\tau$ is used. The weights $\beta_j^*$ depend on $\mathcal{X}$, so it is difficult to compute them in practice when the memory is limited. However, it might be possible to estimate them for some problems.

Nevertheless, with $\beta_j^*$, the above K-prior can achieve perfect reconstruction. The proof is very similar to the one given in Equations 9 and 10, and is shown below,

$$\begin{aligned}
\nabla \mathcal{K}_{\text{opt}}(\mathbf{w}; \mathbf{w}_*, \mathcal{M}^*) &= \sum_{j=1}^K \beta_j^* \mathbf{u}_j^* \left[h(f_{\mathbf{w}}(\mathbf{u}_j^*)) - h(f_{\mathbf{w}_*}(\mathbf{u}_j^*))\right] + \delta(\mathbf{w} - \mathbf{w}_*), \\
&= \mathbf{U}_{1:K}^* \mathbf{D}_u \boldsymbol{\beta}_* + \delta(\mathbf{w} - \mathbf{w}_*), \\
&= \mathbf{U}_{1:K}^* \mathbf{S}_{1:K}^* \mathbf{V}_{1:K}^\top \mathbf{d}_x + \delta(\mathbf{w} - \mathbf{w}_*), \\
&= \mathbf{\Phi}^\top \mathbf{d}_x + \delta(\mathbf{w} - \mathbf{w}_*), \\
&= \sum_{i \in \mathcal{X}} \boldsymbol{\phi}_i \left[h(f_{\mathbf{w}}^i) - h(f_{\mathbf{w}_*}^i)\right] + \delta(\mathbf{w} - \mathbf{w}_*), \\
&= \underbrace{\sum_{i \in \mathcal{D}} \boldsymbol{\phi}_i \left[h(f_{\mathbf{w}}^i) - y_i\right] + \delta\mathbf{w}}_{=\nabla\bar{\ell}(\mathbf{w}).} - \underbrace{\sum_{i \in \mathcal{D}} \boldsymbol{\phi}_i \left[h(f_{\mathbf{w}_*}^i) - y_i\right] - \delta\mathbf{w}_*}_{=0.}.
\end{aligned}$$

The first line is simply the gradient, which is then rearranged in the matrix-vector product in the second line. The third line uses the definition of $\boldsymbol{\beta}_*$, and the fourth line uses the SVD of $\mathbf{\Phi}$. In the fifth line we expand it to show that it is the same as Eq. 9, and the rest follows as before.

Due to their perfect gradient-reconstruction property, we call the prior in Eq. 19 the *optimal* prior. When only top-$M$ singular vectors are chosen, the gradient reconstruction error grows according to the leftover singular values. We show this below where we have chosen $\mathcal{M}_M^* = \{\mathbf{u}_1^*, \mathbf{u}_2^*, \ldots, \mathbf{u}_M^*\}$ as the set of top-$M$ singular vectors,

$$\begin{aligned}
\mathbf{e}_{\text{opt}}(\mathbf{w}; \mathbf{w}, \mathcal{M}_M^*) &= \nabla\bar{\ell}(\mathbf{w}) - \nabla\mathcal{K}_{\text{opt}}(\mathbf{w}; \mathbf{w}_*, \mathcal{M}_M^*) \\
&= \nabla\bar{\ell}(\mathbf{w}) - \nabla\mathcal{K}_{\text{opt}}(\mathbf{w}; \mathbf{w}_*, \mathcal{M}^*) + \nabla\mathcal{K}_{\text{opt}}(\mathbf{w}; \mathbf{w}_*, \mathcal{M}^*) - \nabla\mathcal{K}_{\text{opt}}(\mathbf{w}; \mathbf{w}_*, \mathcal{M}_M^*) \\
&= \nabla\mathcal{K}_{\text{opt}}(\mathbf{w}; \mathbf{w}_*, \mathcal{M}^*) - \nabla\mathcal{K}_{\text{opt}}(\mathbf{w}; \mathbf{w}_*, \mathcal{M}_M^*) \\
&= \sum_{j=M+1}^K \beta_j^* \mathbf{u}_j^* \left[h(f_{\mathbf{w}}(\mathbf{u}_j^*)) - h(f_{\mathbf{w}_*}(\mathbf{u}_j^*))\right], \\
&= \mathbf{U}_{M+1:K}^* \mathbf{S}_{M+1:K}^* \mathbf{V}_{M+1:K}^\top \mathbf{d}_x.
\end{aligned}$$

The first line is simply the definition of the error, and in the second line we add and subtract the optimal K-prior with memory $\mathcal{M}^*$. The next few lines use the definition of the optimal K-prior and rearrange terms.

Using the above expression, we find the following error,

$$\|\mathbf{e}_{\mathrm{opt}}(\mathbf{w}; \mathbf{w}, \mathcal{M}_M^*)\| = \sqrt{\Sigma_{j=M+1}^K s_j^2 (a_j^x)^2}$$

where $a_j^x$ is the $j$'th entry of a vector $\mathbf{a} = \mathbf{V}_{1:K}^\top \mathbf{d}_x$. The error depends on the leftover singular values. The error is likely to be the optimal error achievable by any memory of size $M$, and establishes a theoretical bound on the best possible performance achievable by any K-prior.

# B  Additional Examples of Adaptation with K-priors

Here, we briefly discuss the K-prior regularization for the other adaptation tasks.

## B.1  The Change Regularizer task

For Change Regularizer task, we need to slightly modify the K-prior of Eq. 8. We replace the weight-space divergence in Eq. 8 with a Bregman divergence defined using two different regularizers (see Proposition 5 in Nielsen [42]),

$$\mathcal{B}_{\mathcal{GR}}(\mathbf{w}\|\mathbf{w}_*) = \mathcal{G}(\mathbf{w}) + \mathcal{R}^*(\boldsymbol{\eta}_*) - \mathbf{w}^\top \boldsymbol{\eta}_*, \tag{20}$$

where $\boldsymbol{\eta}_* = \nabla \mathcal{R}(\mathbf{w}_*)$ is the dual parameter and $\mathcal{R}^*$ is the convex-conjugate of $\mathcal{R}$. This is very similar to the standard Bregman divergence but uses two different (convex) generating functions.

To get an intuition, consider hyperparameter-tuning for the $L_2$ regularizer $\mathcal{R}(\mathbf{w}) = \frac{1}{2}\delta\|\mathbf{w}\|^2$, where our new regularizer $\mathcal{G}(\mathbf{w}) = \frac{1}{2}\gamma\|\mathbf{w}\|^2$ uses a hyperparameter $\gamma \neq \delta$. Since the conjugate $\mathcal{R}^*(\boldsymbol{\eta}) = \frac{1}{2}\|\boldsymbol{\eta}\|^2/\delta$ and $\boldsymbol{\eta}_* = \nabla \mathcal{R}(\mathbf{w}_*) = \delta \mathbf{w}_*$, we get

$$\mathcal{B}_{\mathcal{GR}}(\mathbf{w}\|\mathbf{w}_*) = \frac{1}{2}(\gamma\|\mathbf{w}\|^2 + \delta\|\mathbf{w}_*\|^2 - 2\delta\mathbf{w}^\top \mathbf{w}_*).$$

When $\gamma = \delta$, then this reduces to the divergence used in Eq. 8, but otherwise it enables us to reconstruct the gradient of the past objective but with the new regularizer. We define the following K-prior where the weight-divergence is replaced by Eq. 20, and use it to obtain $\hat{\mathbf{w}}_{\mathcal{G}}$,

$$\mathcal{K}(\mathbf{w}; \mathbf{w}_*, \mathcal{M}) = \sum_{i \in \mathcal{M}} \ell\left(h(f_{\mathbf{w}_*}^i), h(f_{\mathbf{w}}^i)\right) + \mathcal{B}_{\mathcal{GR}}(\mathbf{w}\|\mathbf{w}_*),$$
$$\hat{\mathbf{w}}_{\mathcal{G}} = \underset{\mathbf{w} \in \mathcal{W}}{\arg\min} \, \mathcal{K}(\mathbf{w}; \mathbf{w}_*, \mathcal{M}) \tag{21}$$

The following theorem states the recovery of the exact solution.

**Theorem 2.** *For $\mathcal{M} = \mathcal{X}$ and strictly-convex regularizers, we have $\mathbf{w}_{\mathcal{G}} = \hat{\mathbf{w}}_{\mathcal{G}}$.*

The derivation is very similar to Eq. 10, where $\delta(\mathbf{w} - \mathbf{w}_*)$ is replaced by $\nabla \mathcal{G}(\mathbf{w}) - \nabla \mathcal{R}(\mathbf{w}_*)$.

## B.2  The Change Model Class task

We discuss the 'Change Model Class' task through an example. Suppose we want to remove the last feature from $\phi_i$ so that $\mathbf{w} \in \mathbb{R}^P$ is replaced by a smaller weight-vector $\boldsymbol{\theta} \in \mathbb{R}^{P-1}$. Assuming no change in the hyperparameter, we can simply use a weighting matrix to 'kill' the last element of $\mathbf{w}_*$. We define the matrix $\mathbf{A} = \mathbf{I}_{P-1 \times P}$ whose last column is 0 and the rest is the identity matrix of size $P-1$. With this, we can use the following training procedure over a smaller space $\bar{\mathbf{w}}$,

$$\mathcal{K}(\boldsymbol{\theta}) = \sum_{i \in \mathcal{M}} \ell\left(h(f_{\mathbf{w}_*}^i), h(f_{\boldsymbol{\theta}}(\mathbf{x}_i))\right) + \mathcal{B}_{\mathcal{R}}(\boldsymbol{\theta}\|\mathbf{A}\mathbf{w}_*), \qquad \hat{\boldsymbol{\theta}}_* = \underset{\boldsymbol{\theta} \in \Theta}{\arg\min} \, \mathcal{K}(\boldsymbol{\theta}) \tag{22}$$

If the hyperparameters or regularizer are different for the new problem, then the Bregman divergence shown in Eq. 20 can be used, with an appropriate weighting matrix.

Model compression is a specific instance of the 'Change Model Class' task, where the architecture is entirely changed. For neural networks, this also changes the meaning of the weights and the

regularization term may not make sense. In such cases, we can simply use the functional-divergence term in K-priors,

$$\mathcal{K}(\boldsymbol{\theta}) = \sum_{i \in \mathcal{M}} \ell\left(h(f^i_{\mathbf{w}_*}), h(f_{\boldsymbol{\theta}}(\mathbf{x}_i))\right), \qquad \hat{\boldsymbol{\theta}}_* = \arg\min_{\boldsymbol{\theta} \in \Theta} \mathcal{K}(\boldsymbol{\theta}) \tag{23}$$

This is equivalent to knowledge distillation (KD) in Eq. 15 with $\lambda = 0$ and $T = 1$.

Since KD performs well in practice, it is possible to use a similar strategy to boost K-prior, e.g., we can define the following,

$$\hat{\boldsymbol{\theta}}_* = \arg\min_{\boldsymbol{\theta} \in \Theta} \ \lambda \sum_{i \in \mathcal{M}} \ell(y_i, h(f^i_{\boldsymbol{\theta}})) + (1 - \lambda)\mathcal{K}(\boldsymbol{\theta}) \tag{24}$$

We could even use limited-memory in the first term. The term $\lambda$ lets us trade-off teacher predictions with the actual data.

We can construct K-priors to change multiple things at the same time, for example, changing the regularizer, the model class, and adding/removing data. A K-prior for such situations can be constructed using the same principles we have detailed.

## C Derivation of the K-priors Gradients for Deep Learning

The gradient is obtained similarly to (10) where we add and subtract $y_i$ in the first term in the first line below,

$$\nabla\mathcal{K}(\mathbf{w}) = \sum_{i \in \mathcal{X}} \nabla f^i_{\mathbf{w}} \left[ h(f^i_{\mathbf{w}}) - h(f^i_{\mathbf{w}_*}) \right] + \delta(\mathbf{w} - \mathbf{w}_*),$$

$$= \underbrace{\sum_{i \in \mathcal{D}} \nabla f^i_{\mathbf{w}} \left[ h(f^i_{\mathbf{w}}) - y_i \right] + \delta\mathbf{w}}_{=\nabla\ell(\mathbf{w})} - \underbrace{\sum_{i \in \mathcal{D}} \nabla f^i_{\mathbf{w}} [h(f^i_{\mathbf{w}_*}) - y_i] - \delta\mathbf{w}_*}_{\neq\nabla\bar{\ell}(\mathbf{w}_*), \text{ because } \nabla f^i_{\mathbf{w}} \neq \nabla f^i_{\mathbf{w}_*}},$$

The second term is not zero because $\nabla f^i_{\mathbf{w}} \neq \nabla f^i_{\mathbf{w}_*}$ to get $\nabla\bar{\ell}(\mathbf{w}_*)$ in the second term.

The gradient of the KD objective can be obtained in a similar fashion, where we add and subtract $y_i$ in the second term in the first line to get the second line,

$$\nabla\ell_{\text{KD}}(\mathbf{w}) = \lambda \sum_{i \in \mathcal{D}} \nabla f^i_{\mathbf{w}} \left[ h(f^i_{\mathbf{w}}) - y_i \right] + (1 - \lambda) \sum_{i \in \mathcal{D}} \nabla f^i_{\mathbf{w}} \left[ h(f^i_{\mathbf{w}}) - h(f^i_{\mathbf{w}_*}) \right],$$

$$= \sum_{i \in \mathcal{D}} \nabla f^i_{\mathbf{w}} \left[ h(f^i_{\mathbf{w}}) - y_i \right] - (1 - \lambda) \sum_{i \in \mathcal{D}} \nabla f^i_{\mathbf{w}} \left[ h(f^i_{\mathbf{w}_*}) - y_i \right].$$

## D Proof for Adaptation for Bayesian Learning with K-priors

To prove the equivalence of (18) to the full batch variational inference problem with a Gaussian $q(\mathbf{w}) = \mathcal{N}(\mathbf{w}|\boldsymbol{\mu}, \boldsymbol{\Sigma})$, we can use the following fixed point of the variational objective (see Section 3 in [27] for the expression),

$$0 = \nabla_{\boldsymbol{\mu}} \mathbb{E}_q[\mathcal{L}(\mathbf{w})] \big|_{\boldsymbol{\mu}=\boldsymbol{\mu}_+, \boldsymbol{\Sigma}=\boldsymbol{\Sigma}_+} = \mathbb{E}_q[\nabla_{\mathbf{w}}\mathcal{L}(\mathbf{w})]\big|_{\boldsymbol{\mu}=\boldsymbol{\mu}_+, \boldsymbol{\Sigma}=\boldsymbol{\Sigma}_+}, \tag{25}$$

$$\boldsymbol{\Sigma}_+^{-1} = \nabla_{\boldsymbol{\Sigma}} \mathbb{E}_q[\mathcal{L}(\mathbf{w})]\big|_{\boldsymbol{\mu}=\boldsymbol{\mu}_+, \boldsymbol{\Sigma}=\boldsymbol{\Sigma}_+} = \mathbb{E}_q[\nabla_{\mathbf{w}}^2\mathcal{L}(\mathbf{w})]\big|_{\boldsymbol{\mu}=\boldsymbol{\mu}_+, \boldsymbol{\Sigma}=\boldsymbol{\Sigma}_+}, \tag{26}$$

where $\mathcal{L}(\mathbf{w}) = [\ell_j(\mathbf{w}) + \bar{\ell}(\mathbf{w}) + \mathcal{R}(\mathbf{w})]$, $\boldsymbol{\mu}_+$ and $\boldsymbol{\Sigma}_+$ are the mean and covariance of the optimal $q_+(\mathbf{w})$ for the 'Add Data' task. For GLMs, both the gradient and Hessian of $\bar{\ell}(\mathbf{w})$ is equal to those of $\mathcal{K}(\mathbf{w})$ defined in (8), which proves the equivalence.

For equivalence to GPs, we first note that, similarly to the representer theorem, the mean and covariance of $q_+(\mathbf{w})$ can be expressed in terms of the two $N$-length vectors $\boldsymbol{\alpha}$ and $\boldsymbol{\lambda}$ [43, 26, 28],

$$\boldsymbol{\mu}_+ = \boldsymbol{\Phi}_+^\top \boldsymbol{\alpha}, \qquad \boldsymbol{\Sigma}_+ = (\boldsymbol{\Phi}_+^\top \boldsymbol{\Lambda} \boldsymbol{\Phi}_+ + \delta\mathbf{I})^{-1},$$

where $\mathbf{\Lambda}$ is a diagonal matrix with $\boldsymbol{\lambda}$ as the diagonal. Using this, we can define a marginal $q(f_i) = \mathcal{N}(f_i|m_i, v_i)$, where $f_i = \boldsymbol{\phi}_i^\top \mathbf{w}$, with the mean and variance defined as follows,

$$m_i = \boldsymbol{\phi}_i^\top \boldsymbol{\mu}_+ = \mathbf{k}_{i,+}^\top \boldsymbol{\alpha}, \qquad v_i = \boldsymbol{\phi}_i^\top \boldsymbol{\Sigma}_+ \boldsymbol{\phi}_i = k_{ii,+} - \mathbf{k}_{i,+}^\top \left(\mathbf{\Lambda}^{-1} + \delta \mathbf{K}_+\right)^{-1} \mathbf{k}_{i,+},$$

where $k_{ii,+} = \boldsymbol{\phi}_i^\top \boldsymbol{\phi}_i$. Using these, we can now rewrite the optimality conditions in the function-space to show equivalence to GPs.

We show this for the first optimality condition (25),

$$\nabla_{\boldsymbol{\mu}} \mathbb{E}_q[\mathcal{L}(\mathbf{w})]|_{\boldsymbol{\mu}=\boldsymbol{\mu}_+, \boldsymbol{\Sigma}=\boldsymbol{\Sigma}_+} = \sum_{i \in \mathcal{D} \cup j} \mathbb{E}_{\mathcal{N}(\epsilon_i|0,1)} \left[\nabla_f \ell(y_i, h(f))|_{f=\boldsymbol{\phi}_i^\top \boldsymbol{\mu}_+ + \left(\boldsymbol{\phi}_i^\top \boldsymbol{\Sigma}_+ \boldsymbol{\phi}_i\right)^{1/2}\epsilon_i}\right] \boldsymbol{\phi}_i + \delta \boldsymbol{\mu}_+$$

Multiplying it by $\boldsymbol{\Phi}_+$, we can rewrite the gradient in the function space,

$$0 = \sum_{i \in \mathcal{D} \cup j} \mathbb{E}_{\mathcal{N}(\epsilon_i|0,1)} \left[\nabla_f \ell(y_i, h(f))|_{f=m_i+v_i^{1/2}\epsilon_i}\right] \mathbf{k}_{i,+} + \delta \mathbf{K}_+ \boldsymbol{\alpha}$$

$$= \sum_{i \in \mathcal{D} \cup j} \nabla_{m_i} \mathbb{E}_{q(f_i)} \left[\ell(y_i, h(f_i))\right] \mathbf{k}_{i,+} + \delta \mathbf{K}_+ \boldsymbol{\alpha}$$

where $\mathbf{m}$ is the vector of $m_i$. Setting this to 0, gives us the first-order condition for a GP with respect to the mean, e.g., see Equation 3.6 and 4.1 in Chapelle [14]. It is easy to check this for GP regression, where $\ell(y_i, h(f_i)) = (y_i - f_i)^2$, in which case, the equation becomes,

$$0 = \sum_{i \in \mathcal{D} \cup j} (m_i - y_i)\mathbf{k}_{i,+} + \delta \mathbf{K}_+ \boldsymbol{\alpha} \quad \Rightarrow \boldsymbol{\alpha} = (\mathbf{K}_+ + \delta \mathbf{I})^{-1}\mathbf{y},$$

which is the quantity which gives us the posterior mean. A similar condition condition for the covariance can be written as well.

Clearly, when we use a limited memory, some of the data examples are removed and we get a sparse approximation similarly to approaches such as informative vector machine which uses a subset of data to build a sparse approximation [23]. Better sparse approximations can be built by carefully designing the functional divergence term. For example, we can choose the matrix $\mathbf{B}$ in the divergence,

$$\mathbb{D}_f(\mathbf{f}(\mathbf{w})\|\mathbf{f}(\mathbf{w}_*)) = \tfrac{1}{2}\mathbf{d}_m^\top \mathbf{B}\mathbf{d}_m \quad \Rightarrow \quad \nabla \mathbb{D}_f(\mathbf{f}(\mathbf{w})\|\mathbf{f}(\mathbf{w}_*)) = \nabla \mathbf{f}(\mathbf{w})^\top \mathbf{B}\mathbf{d}_m$$

This type of divergence is used in Pan et al. [45], where the matrix $\mathbf{B}$ is set to correlate the examples in $\mathcal{M}$ with the examples in $\mathcal{D}$. Design of such divergence function is a topic which requires more investigation in the future.

# E  Further experimental results

We provide more details on all our experiments, such as hyperparameters and more results.

## E.1  Adaptation tasks

**Logistic Regression on the 'UCI Adult' dataset.** In Fig. 2(a) we show results for the 4 adaptation tasks on the UCI Adult dataset, and provide experimental details in Sec. 5. Note that for all but the 'Change Model Class' task, we used polynomial degree 1. For all but the 'Change Regularizer' task, we use $\delta = 5$.

We optimize using LBFGS (default PyTorch implementation) with a learning rate of $0.01$ until convergence. Throughout our experiments in the paper, we used the same memorable points for Replay as for K-priors (the points with the highest $h'(f_{\mathbf{w}_*}^i)$), and used $\tau = 1$ (from Eq. 6). In Fig. 4 we provide an ablation study for Replay with different strategies: (i) we choose points by $h'(f_{\mathbf{w}_*}^i)$ and use $\tau = N/M$, (ii) we choose points randomly and use $\tau = 1$, (iii) we choose points randomly and use $\tau = N/M$. Recall that $N$ is the past data size (the size of $\mathcal{D}$) and $M$ is the number of datapoints stored in memory (the size of $\mathcal{M}$). We see that choosing points by $h'(f_{\mathbf{w}_*}^i)$ and using $\tau = 1$ performs very well, and we therefore choose this for all our experiments.

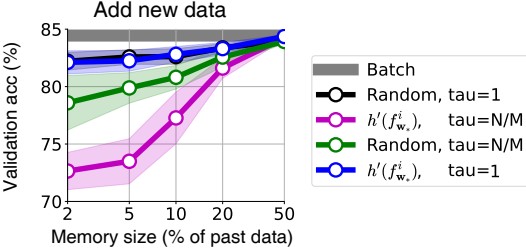

Figure 4: This figure shows using $\tau = 1$ works well for Replay, both for random selection of memory and choosing memory by sorting $h'(f^i_{\mathbf{w}_*})$. We compare different methods for Replay on the Adult 'Add Data' task. 'Random' means the points in memory are chosen randomly as opposed to choosing the points with highest $h'(f^i_{\mathbf{w}_*})$. We also consider using $\tau = N/M$ instead of $\tau = 1$. Choosing randomly or by $h'(f^i_{\mathbf{w}_*})$ are within standard deviations in this task, so we choose to report memory chosen by $h'(f^i_{\mathbf{w}_*})$ in other experiments (this is then consistent with the memory in K-priors).

**Logistic Regression on the 'USPS odd vs even' dataset.** For all but the 'Change Model Class' task, we used polynomial degree 1. For all but the 'Change Regularizer' task, we use $\delta = 50$. We optimize using LBFGS with a learning rate of $0.1$ until convergence.

**Neural Networks on the 'USPS odd vs even' dataset.** For all but the 'Change Regularizer' task, we use $\delta = 5$. We optimize using Adam with a learning rate of $0.005$ for 1000 epochs (which is long enough to reach convergence).

**Neural Networks on the 'MNIST' dataset.** We show results on 10-way classification with MNIST in Fig. 5, which has 60,000 training images across 10 classes (handwritten digits), with each image of size $28 \times 28$. We use a two hidden-layer MLP with 100 units per layer, and report means and standard deviations across 3 runs. For the 'Add Data' task, we start with a random 90% of the dataset and add 10%. For the 'Change Regularizer' task, we change $\delta = 1$ to 5 (we use $\delta = 1$ for all other tasks). For the 'Change Architecture' task, we compress to a single hidden layer with 100 hidden units. We optimize using Adam with a learning rate of $0.001$ for 250 epochs, using a minibatch size of $512$.

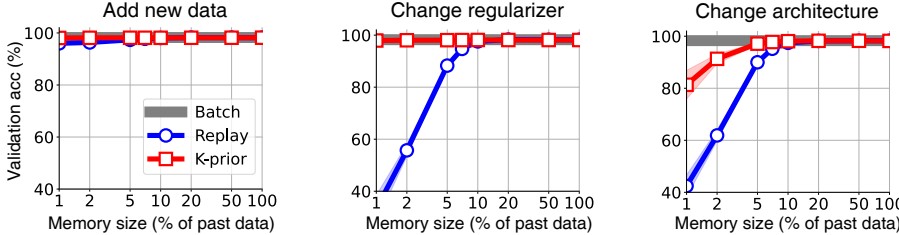

Figure 5: K-priors work well on MNIST (with an MLP), similar to other results on the USPS and UCI Adult datasets. For details on the experiments, see App. E.1.

**Neural Networks on the 'CIFAR-10' dataset.** We provide results for CIFAR-10 using 10-way classification. CIFAR-10 has 60,000 images (50,000 for training), and each image has 3 channels, each of size $32 \times 32$. We report mean and standard deviations over 3 runs. We use the CifarNet architecture from Zenke et al. [62]. We optimize using Adam with a learning rate of $0.001$ for 100 epochs, using a batch size of $128$.

In Fig. 6 we also provide results on the 'Change Regularizer' task, where we change $\delta = 1$ to $0.5$ (we use $\delta = 1$ for all the other tasks). We also provide results on the 'Change Architecture' task, where we change from the CifarNet architecture to a LeNet5-style architecture. This smaller architecture has two convolution layers followed by two fully-connected layers: the first convolution layer has 6 output channels and kernel size 5, followed by the ReLU activation, followed by a Max Pool layer with kernel size 2 (and stride 2), followed by the second convolution layer with 16 output channels and kernel size 5, followed by the ReLU activation, followed by another Max Pool layer with kernel size 2 (and stride 2), followed by a fully-connected layer with 120 hidden units, followed

by the last fully-connected layer with 84 hidden units. We also use ReLU activation functions in the fully-connected layers.

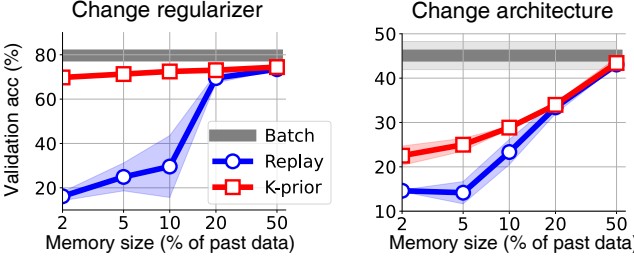

Figure 6: Results for two adaptation tasks on CIFAR-10 with CNNs. See also Fig. 3(c) for results on the 'Add Data' task. K-priors perform well, especially on the 'Change Regularizer' task. The 'Change Architecture' task is more difficult, but we note that we do not use a temperature. Having a temperature greater than 1 is known to help in similar settings, such as knowledge distillation [24].

For the knowledge distillation task, we used K-priors with a temperature, similar to the temperature commonly used in knowledge distillation [24]. We note that there is some disagreement in the literature regarding how the temperature should be applied, with some works using a temperature only on the teacher's logits (such as in Eq. 15) [37], and other works having a temperature on both the teacher and student's logits [24]. In our experiments, we use a temperature $T$ on both the student and teacher logits, as written in the final term of Eq. 27. We also multiply the final term by $T^2$ so that the gradient has the same magnitude as the other data term (as is common in knowledge distillation).

$$\ell_{\mathrm{KD,expt}}(\mathbf{w}) = \lambda \sum_{i \in \mathcal{D}} \ell\left(y_i, h(f_{\mathbf{w}}^i)\right) + \delta \|\mathbf{w}\|^2 + (1 - \lambda) T^2 \sum_{i \in \mathcal{D}} \ell\left(h(f_{\mathbf{w}_*}^i/T), h(f_{\mathbf{w}}^i/T)\right). \quad (27)$$

We used $\lambda = 0.5$ in the experiment. We performed a hyperparameter sweep for the temperature (across $T = [1, 5, 10, 20]$), and used $T = 5$. For K-priors in this experiment, we optimize for 10 epochs instead of 100 epochs, and use $\tau = 1$.

In Fig. 3(c) we also showed initial results using a temperature on the 'Add Data' task on CIFAR-10. We used the same temperature from the knowledge distillation experiment ($T = 5$ and $\lambda = 0.5$), but did not perform an additional hyperparameter sweep. We find that using a temperature improved results for CNNs, and we expect increased improvements if we perform further hyperparameter tuning. Note that many papers that use knowledge distillation perform more extensive hyperparameter sweeps than we have here.

### E.2 Weight-priors vs K-priors

In Fig. 7 we provide results comparing with weight-priors for all the 'Add Data' tasks. We see that for homogeneous data splits (such as UCI Adult, MNIST and CIFAR), weight-priors perform relatively well. For inhomogeneous data splits (USPS with logistic regression and USPS with neural networks), weight-priors perform worse.

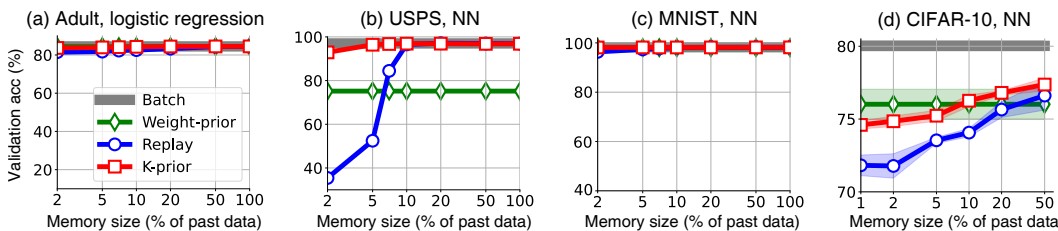

Figure 7: Results on the 'Add Data' task, with a comparison to weight-priors. (a), (c), (d) For homogeneous data splits, weight-priors can perform relatively well. (b) For inhomogeneous data splits, weight-priors perform worse (see also Fig. 3(b)).

### E.3 K-priors ablation with weight-term

In this section we perform an ablation study on the importance of the weight-term $\frac{1}{2}\delta\|\mathbf{w} - \mathbf{w}_*\|^2$ in Eq. 8. In Fig. 8 we show results on logistic regression on USPS where we do not have $\mathbf{w}_*$ in this term (the update equation is the same as Eq. 8 except the weight-term is $\frac{1}{2}\delta\|\mathbf{w}\|^2$ instead of $\frac{1}{2}\delta\|\mathbf{w} - \mathbf{w}_*\|^2$). We see that the weight-term is important: including the weight-term always improves performance.

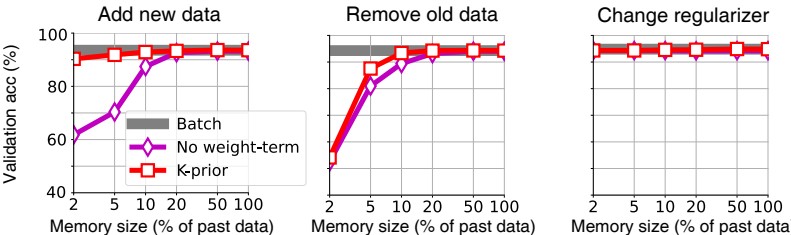

Figure 8: Comparing K-priors with a version of K-priors without the weight-term on USPS logistic regression. We see that the weight-term is important, especially on the 'Add Data' task.

### E.4 K-priors with random initialization

In all experiments so far, when we train on a new task, we initialize the parameters at the previous parameters $\mathbf{w}_*$. Note that this is not possible in the "Change architecture" task, where weights were initialized randomly. Our results are independent of initialization strategy: we get the same results whether we use random initialization or initializing at previous values. The only difference is that random initialization can sometimes take longer until convergence (for all methods: Batch, Replay and K-priors).

For GLMs, where we always train until convergence and there is a single optimum, it is clear that the exact same solution will always be reached. We now also provide the result for 'USPS odd vs even', with random initialization in Fig. 9, for the 3 tasks where we had earlier initialized at previous values (compare with Fig. 1 (right)). We use exactly the same hyperparameters and settings as in Fig. 1 (right), aside from initialization method.

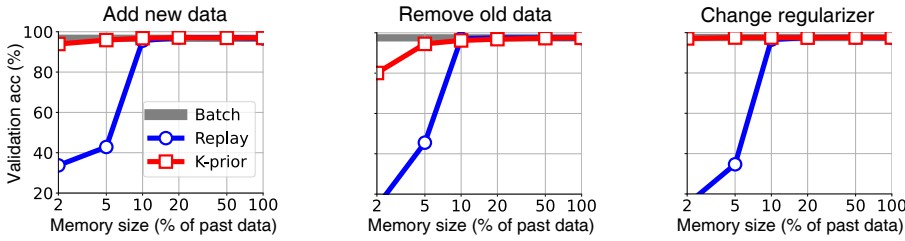

Figure 9: K-priors obtain the same results when randomly initializing the weights for the 'Add new data', 'Remove old data' and 'Change regularizer' tasks on USPS odd vs even with neural networks. Previous results, including Fig. 1 (right), initialized parameters at previously learnt values. The 'Change architecture' task originally used random initialization and so is not repeated here.

### E.5 K-priors converge cheaply

In this section, we show that K-priors with limited memory converge to the final solution cheaply, converging in far fewer passes through data than the batch solution. This is because we use a limited memory, and only touch the more important datapoints.

Table 1 shows the "number of backprops" until reaching specific accuracies (90% and 97%) on USPS with a neural network (using the same settings as in Fig. 1 (right)). This is one way of measuring the "time taken", as backprops through the model are the time-limiting step. For K-priors and Replay, we use 10% of past memory. All methods use random initializations when starting training on a new task.

We see that K-priors with 10% of past data stored are quicker to converge than Batch, even though both eventually converge to the same accuracy (as seen in Fig. 1 (right)). For example, to reach 97% accuracy for the Change Regularizer task, K-priors only need 54,000 backward passes, while Batch requires 2,700,000 backward passes. We also see that Replay is usually very slow to converge. This is because it does not use the same information as K-priors (as Replay uses hard labels), and therefore requires significantly more passes through data to achieve the same accuracy. In addition, Replay with 10% of past data cannot achieve high accuracies (such as 97% accuracy), as seen in Fig. 1 (right).

Table 1: Number of backpropagations required to achieve a specified accuracy on USPS with a neural network (1000s of backprops). K-priors with 10% past memory require much fewer backprops to achieve the same accuracy as Batch, while Replay with 10% memory cannot achieve high accuracies.

| Accuracy achieved | Method | Add new data | Remove old data | Change regularizer | Change model class |
|---|---|---|---|---|---|
| 90% | Batch | 87 | 94 | 94 | 86 |
| 90% | Replay (10% memory) | 348 | 108 | 236 | 75 |
| 90% | **K-prior (10% memory)** | **73** | **53** | **13** | **22** |
| 97% | Batch | 1,900 | 1,800 | 2,700 | 3,124 |
| 97% | Replay (10% memory) | – | 340 | – | – |
| 97% | **K-prior (10% memory)** | **330** | **120** | **54** | **68** |

### E.6  Further details on Fig. 1 (middle), moons dataset.

To create this dataset, we took 500 samples from the moons dataset, and split them into 5 splits of 100 datapoints each, with each split having 50 datapoints from each task. Additionally, the splits were ordered according to the x-axis, meaning the 1st split were the left-most points, and the 5th split had the right-most points. In the provided visualisations, we show transfer from 'past data' consisting of the first 3 splits (so, 300 datapoints) and the 'new data' consisting of the 4th split (a new 100 datapoints). We store 3% of past data as past memory in K-priors, chosen as the points with the highest $h'(f_{\mathbf{w}_*}^i)$.

## F   Changes in the camera-ready version compared to the submitted version

This section lists the major changes we made for the camera-ready version of the paper, incorporating reviewer feedback.

- Added a paragraph on the optimal K-prior after Eq. 12, as well as a detailed explanation in App. A.
- Updated Fig. 3(d), following a more extensive sweep of hyperparameters.
- Added App. E.4, showing K-priors with random initialization give the same results as K-priors that are initialized at the previous model parameters.
- Added App. E.5, showing that K-priors with limited memory converge to the final solution cheaply, requiring fewer passes through the data than the batch solution.