# OpenReview forum: "Knowledge-Adaptation Priors"
_NeurIPS.cc/2021/Conference — NeurIPS 2021 Poster_

### Official Review · Reviewer_AAdt · 2021-07-13

**Rating:** 6
**Confidence:** 4

**Summary:**

The paper proposes K-priors, giving a family of approaches aiming to quickly update trained networks to tasks changed in various ways (with added data, removed data, a changed regularizer, or a changed architecture). The main features of K-priors are: they approximate (or in some cases compute exactly) gradients of the original objective using the outputs of a trained model instead of the data labels; and they can approximate the original objective well with only a small fraction of the original data.

**Limitations And Societal Impact:**

The authors claim, I think reasonably, that this work has little obvious potential for negative societal impact.

EDIT: Removed sentence regarding concerns about reduction in training time.

**Main Review:**

# Novelty
The difference between the K-priors objective in Eq. 10 and full retraining is fairly small - as stated in line 130, it is that model outputs (with old weights) $h(f_{w^*}^i)$ are used in place of data labels $y_i$. This is similar to Knowledge Distillation [22] (cited in paper), as noted in the paper. The "memorable past examples" approach of [37] (cited in paper) is combined with this to reduce memory requirements - as far as I know, such a combination is novel.

# Strengths
- The paper is well-motivated, and I believe that reducing the cost and environmental impact of retraining models is an important line of work.
- Equation 9 is an interesting result, showing that their approach exactly recovers the original gradients for generalized linear models.
- The experimental results show significant improvements with using K-priors over 'Replay' when only a small amount of past data is kept.

# Weaknesses
- (EDIT: The authors have presented results satisfying me that updating weights using K-priors can indeed be significantly faster than retraining.) A major aspect of the motivation for this work is to reduce the time/energy consumption for retraining, but this is not demonstrated anywhere in the experiments (only a reduction in the number of examples that need to be stored in memory is demonstrated). If this is not to be included then, at the very least, the abstract should not claim that the empirical results "confirm that the adaptation can be cheap and accurate". To me, "cheap" implies that the training time/energy consumption is reduced.
- It is difficult to interpet the performance on the "Remove old data" task by looking at the plots of validation accuracy - surely very good validation accuracy could be achieved by simply keeping the old weights and not removing any data.
- Despite the response to 3a on the checklist, there does not seem to be any code included with the submission.

# Writing
- The writing is mostly of a good standard, but I think the novel contributions should be more clearly stated.
- When performing adaptation, could the authors confirm whether weights are randomly initialised or initialised from the previously learned values?
- The introduction of Eq. 4 is confusing - is $y$ a scalar, a one-hot encoding, or other vector? Similarly for $f$, is it a scalar, vector, etc. (i.e. what is $\mathcal{F}$)?
- (EDIT: I am no longer concerned about this point, and now understand that the objective suggested for each task are unified by being something like "the simplest objectives which recover/approximate the gradients of the desired new objective without using data labels".) It is a little unclear what unifies the different objectives presented as K-priors - e.g. Equation 10 for adding data, in which $\mathcal{K}(w)$ has the same gradients as the old objective, and Equation 12 for changing the regulariser, in which $\mathcal{K}(w)$ is modified based  on the new regulariser, and then used on its own as the new objective. Then in equation 19 for changing the model class, $\mathcal{K}(\theta)$ is used as the new objective, and no longer contains a weight-space divergence. It would be helpful if the authors could be clearer about that unifies these, and distinguishes them from prior work.

EDIT SUMMARY: See bullet point 1 under "Weaknesses" and bullet point 4 under "Writing".

**Time Spent Reviewing:**

8

---

> ### Author Response · Authors · 2021-08-10
> **Response to Reviewer AAdt**
>
> Q1: If this is not to be included then, at the very least, the abstract should not claim that the empirical results "confirm that the adaptation can be cheap and accurate". To me, "cheap" implies that the training time/energy consumption is reduced.
>
> A: We will clarify what we mean by “cheap”. Our focus is on reducing the cost *and* frequency of “full” retraining. This is mentioned in the first paragraph of introduction, where in line 18 we say that “the cost can be reduced if, instead of repeatedly retraining from scratch, the system can quickly adapt to small changes.”
>
> Q2: Despite the response to 3a on the checklist, there does not seem to be any code included with the submission.
>
> A: Apologies, we will change the response to 3a on the checklist to be “No”. We were earlier confused by the wording, we mean to say yes to “instructions needed to reproduce the main experimental results”, but we have not provided code, and realize now that this is a mistake.
>
> Q3: When performing adaptation, could the authors confirm whether weights are randomly initialised or initialised from the previously learned values?
>
> A: In all experiments, weights are initialized from the previously learned values (we will clarify this in the text), as it was quicker to train. We will add results with random initialization in an appendix. Results are the same (within error) as with initializing at previous values. Note that for GLMs, where there is a global optimal point, initialization strategy does not affect results.
>
> Q4: It is difficult to interpret the performance on the "Remove old data" task by looking at the plots of validation accuracy - surely very good validation accuracy could be achieved by simply keeping the old weights and not removing any data.
>
> A: As we say above, we can initialise the weights randomly (instead of initialising from previously learned values), and we still obtain the same results. This shows that the model is not simply using the old weight values to maintain good validation accuracy.
>
> Q5: The writing is mostly of a good standard, but I think the novel contributions should be more clearly stated.
>
> A: We will improve this by rewriting parts of the introduction.
>
> Q6: The introduction of Eq. 4 is confusing - is a scalar, a one-hot encoding, or other vector? Similarly for , is it a scalar, vector, etc. (i.e. what is )?
>
> A: The definition of y is a standard one as used to define the GLM. We will make the distinction to the vector case clearer in writing.
>
> Q7: It is a little unclear what unifies the different objectives presented as K-priors - e.g. Equation 10 for adding data, in which has the same gradients as the old objective, and Equation 12 for changing the regulariser, in which is modified based on the new regulariser, and then used on its own as the new objective. Then in equation 19 for changing the model class, is used as the new objective, and no longer contains a weight-space divergence. It would be helpful if the authors could be clearer about that unifies these, and distinguishes them from prior work.
>
> A: This appears to be a misunderstanding.
> - Whenever new data is added/removed, we add the corresponding loss terms (eg. Eq. 10). Changing regularizer/architecture does not add any new data, so there are no data loss-terms in the objective (Eq. 12 and 19).
> - The choice of divergence function changes depending on the task. For the Change Regularizer task, the weight-divergence needs to swap the regularizers, that’s why it’s different.
>
> We will clarify this in the paper.

---

> > ### Comment · Reviewer_AAdt · 2021-08-17
> > **Thanks for the response**
> >
> > ### Q1
> > My understanding is that, when the desired training setup changes, you suggest performing a fast adaptation of the model by optimising one of the objectives in Equation 2 or 3 and (given your response to Q3) starting from the previously learned weights. Given that you said you do initialise from the previous weights, I would expect that this is faster than performing full retraining whenever the desired training setup changes, and so my concerns are somewhat alleviated (although I have little intuition for how much this initialisation helps with e.g. logistic regression). I would still be interested to see quantitatively how much faster this is (compared to full retraining) in the experiments reported.
> >
> > ### Q7
> > Thanks for your reply. I had understood these points. What was unclear to me was the reasoning behind the form chosen for $\mathcal{K}(w)$ in each case, and whether there was a unifying framework behind it. After spending some more time looking into this, I am satisfied that they are the simplest forms that satisfy the principles described in the paragraph "A principle for constructing K-priors".

---

> > ### Comment · Reviewer_AAdt · 2021-08-23
> > **Remaining concern**
> >
> > Let me be clear about my main remaining concern: I am not sure whether (or by how much) your adaptation is actually cheaper than full retraining, given that absolutely no results on training time are given in the paper or appendix. This concern was not addressed by your response to Q1. It may be addressed by e.g. a plot similar to Figure 2 or Figure 3b-d with "Training time" (measured in some suitable way) rather than "Validation acc." on the y-axis. Or alternatively, by a plot similar to Figure 2 or Figure 3b-d with "Memory size" replaced by "Number of training/adaptation iterations" on the x-axis (and memory size kept fixed).
> >
> > I am now generally happy with the remainder of the paper, and would recommend acceptance if you can address this adequately.

---

> > > ### Author Response · Authors · 2021-08-26
> > > **Response to Reviewer AAdt**
> > >
> > > Thanks for your responses. We are glad that many of your concerns have been alleviated, and thank you for explicitly explaining your remaining concern.
> > >
> > > In the table below, we show what you request. The table shows the “number of backprops” until reaching specific accuracies (90% and 97%) on USPS with a neural network (we use the same settings as in Figure 1 (right)). This is one way of measuring the “time taken” as you mentioned, as backprops through the model are the time-limiting step. For K-priors, we use 10% of past memory. Both K-priors and batch use random initializations, so we are truly measuring time taken to a specified accuracy.
> > >
> > > The result shows that K-priors with 10% of past data stored are quicker to converge than batch, even though both eventually converge to the same accuracy (as seen in Figure 1 (right)). For example, to reach 97% accuracy for the Change Regularizer task, K-priors only need 54,000 backward passes, while Batch requires 2,700,000 backward passes.
> > >
> > > Table: Number of backprops required to achieve a specified accuracy on USPS with a neural network (1000s of backprops)
> > >
> > > | Accuracy achieved |  Method  |  Add new data  |  Remove old data  |  Change regularizer  |  Change model class  |
> > > |:----:|:---:|:-----:|:----:|:----:|:-----:|
> > > | 90% | Batch |  87 |  94 | 94 | 86 |
> > > | 90% | K-prior (10% memory) | 73 | 53 | 13 | 22 |
> > > | 97% | Batch |  1,900 |  1,800 | 2,700 | 3,100 |
> > > | 97% | K-prior (10% memory) | 330 | 54 | 330 | 68 |

---

> > > > ### Comment · Reviewer_AAdt · 2021-08-27
> > > > **Thanks for the results**
> > > >
> > > > Thank you for presenting these results showing that updates with K-priors are faster than full retraining. I've updated my original review and score. I hope you will include these results and discussion on the resulting speed-up given by using K-priors in the updated manuscript, as this seems necessary to evaluate how practically useful the method is. I'd also be interested to see results on (but, to be clear, am not asking for any new results in this discussion period) why this time  (or number of backprops) is so much lower with K-priors. E.g. is this due to the small % of memory used by K-priors, or is it more due to a "stronger" learning signal provided by using model outputs $f_{\mathbf{w}}^i$ instead of the data labels?

---

> > > > > ### Author Response · Authors · 2021-08-30
> > > > > **Thanks for increasing the score**
> > > > >
> > > > > We thank you for increasing the score. We promise to include the result in the paper. You are right: the number of backprops are lower due to the small % of memory used by K-priors. Also note that, by using the model outputs rather than labels, we get more out of those data points and K-priors give better results than Replay. In the final version, we will include comparisons to both "batch" and "Replay" vs the "number of backprops".
> > > > >
> > > > > Thanks again for taking your time and feedback. We greatly appreciate your efforts!

---

### Official Review · Reviewer_wqo5 · 2021-07-14

**Rating:** 7
**Confidence:** 3

**Summary:**

This paper proposes a new regularizer to adapt the prior knowledge through weight prior and the memory that is smaller than the entire dataset but identical to be learned correctly. Previous methods to do this have focused on the "Add/Remove Data" task, but the authors extended the adaptation tasks to adapt regularizer change or architecture change through the proposed regularizer, K-priors. It also could be applied to a variety of models from a linear model to deep learning. They described the connections between K-priors and previous works for adding/removing data for SVMs, knowledge distillation and continual learning for deep learning, and Bayesian learning. They mainly validated that K-priors outperform the model without K-priors but using the same memory selected through K-priors for logistic regression and classification tasks.

**Limitations And Societal Impact:**

As I mentioned in the previous section, stability could be a limitation of this research, but I am not sure.

When applying it to industrial or real applications, it sustains some samples as a memory, the right to be forgotten for the information on the internet may be infringed.

**Main Review:**

The authors tried to contribute the unified regularizer that is based on the previous knowledge (when there is no previous knowledge, it would be used like$l_2-\text{norm}$ weight regularizer). The previous knowledge exists as samples or trained parameters. They tried to use the two type knowledge with their prior. This paper has several strengths, one of them is that they tried to design the general prior that is not limited in deep learning, which is rare in those days (I don't think research on deep learning is not good, but diversity on research will make our community healthy more). Another is that they described the connections with previous works like knowledge distillation. However, the validations are on small size of experimental setting, and I am not sure about the stability of this prior. Another concern is about forgetting issue when updating the model multiple times. The memory is reset in every time, then the identical samples that are stored in memory will be removed and the model can forget that. However, it could be overcame through diving the entire memory to a sub-memory for each update I think. I agree too that the experiment for CIFAR-10 knowledge distillation (the dip in the middle in Fig 3 (d)) must be updated or analyzed more.

Questions
- In line 64-65, how $k$-fold Cross-validation is related with adaptation task?


**Time Spent Reviewing:**

10 Hours

---

> ### Author Response · Authors · 2021-08-10
> **Response to Reviewer wqo5**
>
> Q1: The validations are on small size of experimental setting, and I am not sure about the stability of this prior.
>
> A: We are not sure we understand the concern about the “stability” of the prior. Could the reviewer please provide further detail here? Regarding the “small size” comment, our experiments do include some medium sized problems too (e.g., CIFAR-10 and CifarNet).
>
> Q2: Another concern is about forgetting issue when updating the model multiple times. The memory is reset in every time, then the identical samples that are stored in memory will be removed and the model can forget that. However, it could be overcame through diving the entire memory to a sub-memory for each update I think.
>
> A: Regarding “updating the model multiple times”, we mention in line 155 that K-priors is similar to the FROMP method of [37]. The memory can be managed in a similar fashion as that method which has been shown to work well for continual learning to avoid forgetting.
>
> Q3: I agree too that the experiment for CIFAR-10 knowledge distillation (the dip in the middle in Fig 3 (d)) must be updated or analyzed more.
>
> A: We have significantly improved the CIFAR-10 knowledge distillation results (Figure 3(d)). Now, we achieve performance close to full knowledge distillation at all small memory sizes. Using 10% of past memory, the validation accuracy is $59.4 \pm 0.5$%, and using 1% of memory has a validation accuracy of $60.1 \pm 1.2$% (mean and standard deviation over 3 runs; we will update Figure 3(d)). Note that the previously observed dip is now within error. We achieved this improvement by reducing $\lambda$ by an order of magnitude, while keeping all other hyperparameters the same (we note that knowledge distillation in practice always uses similarly small $\lambda$ values, and we earlier hypothesized that doing so would also improve our results in Appendix D.1 lines 695-697).
>
> Q4: How k-fold Cross-validation is related with adaptation task?
>
> A: In k-fold CV, we train multiple models, each with slightly different training settings. For example, consider a 4-fold CV, with data partitions (A,B,C,D). Suppose we first leave out the partition D and train the model on (A,B,C) for a given hyperparameter. Then, we can use this model to “warm-start” the next fold’s training, say (B,C,D), which corresponds to Add Data (partition D) and Remove Data (partition A) tasks. We can continue in this fashion. See Fig. 1 in [52] for an example. We will add more explanation in the paper.
>
> [52] Zeyi Wen, Bin Li, Ramamohanarao Kotagiri, Jian Chen, Yawen Chen, and Rui Zhang. Improving efficiency of svm k-fold cross-validation by alpha seeding. In Proceedings of the AAAI Conference on Artificial Intelligence, volume 31, 2017.
>
> Q5: When applying it to industrial or real applications, it sustains some samples as a memory, the right to be forgotten for the information on the internet may be infringed.
>
> A: It is important to note that the memory is implicit, hidden somewhere inside the model (e.g. in model weights). K-priors add an explicit memory to make it easier to access the implicit memory. This way it has the potential to improve privacy and make it easier to forget without compromising the accuracy. We agree that this setup requires modifications according to existing law, or vice versa, which requires more work in the future.

---

> > ### Comment · Reviewer_wqo5 · 2021-08-10
> > **Fixing typo**
> >
> > Thank you for your response, I made a typo, stability -> scalability (The chrome automatically changed it :p)

---

> > > ### Author Response · Authors · 2021-08-12
> > > **Thanks for the clarification**
> > >
> > > Thanks for the clarification!
> > >
> > > Scalability is not a problem because K-prior can be trained with a first-order method (e.g., SGD), similarly to a full retraining-from-scratch. The main difference is that now we only use a subset of input locations with model’s predictions (instead of the true labels). Predictions can be obtained with a forward pass (which is already computed during backprop), so there is no extra costs involved.

---

### Official Review · Reviewer_5dEe · 2021-07-15

**Rating:** 4
**Confidence:** 3

**Summary:**

This paper proposes a method called K-priors for adapting a model to newly coming data,  new regularization, or new architecture.  It suggests such adaption can be done by minimizing the Bregman divergence between features generated by previous model and current model,  as well as minimizing the Bregman divergence between parameters of previous model and current model.

**Limitations And Societal Impact:**

Limitations are stated above. No societal impact in my understanding.

**Main Review:**

The novelty is incremental, as the form of Eq.7 is just a combination of knowledge distillation loss and a weight regularization, both of which are common in continual learning literature.
And the claims are not supported by the definition of the method.  According to Eq.9, only when M=D the gradient can recover the gradient of previous loss, which is not surprising at all. However, authors claim that data out of the previous distribution can be used to approximate the previous gradient (Line 133), how can this be guaranteed?
Also, the suggested way to select memorized samples is cumbersome and there are existing approaches proposed for sample selection in continual learning as well, e.g. [1],[2].  The authors didn't discuss or compare with any such related work.  And how far this approximation can be to the previous gradient is not analyzied either. So, how to achieve the claim of 'recover the exact retrained model to an arbitrary accuracy by choosing a sufficiently large memory of the past data'?  For a specified accuracy, how large the memory should be?

[1]. Aljundi, R., Belilovsky, E., Tuytelaars, T., Charlin, L., Caccia, M., Lin, M., and Page-Caccia, L. Online Continual Learning with Maximal Interfered Retrieval. In Advances in Neural Information Processing Systems, pp. 11849–11860, 2019a.
[2]. Aljundi, R., Lin, M., Goujaud, B., and Bengio, Y. Gradient-based Sample Selection for Online Continual Learning. In Advances in Neural Information Processing Systems, pp. 11816–11825, 2019b.

**Time Spent Reviewing:**

3 hours

---

> ### Author Response · Authors · 2021-08-10
> **Response to Reviewer 5dEe**
>
> Q1: The novelty is incremental, as the form of Eq.7 is just a combination of knowledge distillation loss and a weight regularization, both of which are common in continual learning literature.
>
> A: This is incorrect. Our main contribution is to show that such a combination works because it reconstructs the past gradients faithfully. No earlier work in continual learning has shown this before (see lines 247-263 for a discussion). Also, no earlier works have used this combination for the “change regularizer” task.
>
> The other reviewers agree with us too. For example, see the comment by Reviewer 2Zuk, where they state that “training on the K-prior objective on the entire dataset recovers the original model - which would *not* be true if we only have weight space regularization, or only have prediction regularization.” (https://openreview.net/forum?id=_cXX-Dr7sf0&noteId=z-jD7RU3LR)
>
> Q2: The claims are not supported by the definition of the method. According to Eq.9, only when M=D the gradient can recover the gradient of previous loss, which is not surprising at all.
>
> A: We strongly disagree. The result is surprising since, using K-priors, we can approximation the gradients without requiring any labels, and just from the *model predictions*. All the other reviewers agree with this point (specifically see Q4 by Reviewer 2Zuk https://openreview.net/forum?id=_cXX-Dr7sf0&noteId=v2b6kL4N4f-).
>
> Q3: However, authors claim that data out of the previous distribution can be used to approximate the previous gradient (Line 133), how can this be guaranteed?
>
> A: This is fairly straightforward, but due to the space limit, we could not include the details (we mention in line 196 the use of “singular-value decomposition (SVD)” to show this). Essentially, in Eq. 8, the first term can be replaced by input locations $u_i$ which are the singular vector of $\Phi$ (and not part of the training data). This is shown below:
>
> $\sum_{i\in\mathcal{D}} \phi_i  [ h(f_w^i) - h(f_{w_*}^i)] =  \sum_{j=1}^K u_j \beta_j [ h(f_w(u_j)) - h( f_{w_*}(u_j)) ]$
>
> where $u_i$ are left singular-vector of the SVD $\Phi^\top = U_{1:K} S_{1:K} V_{1:K}^\top$ with $K$ being the rank of $\Phi$, and
>
> $\beta = \text{Diag}(d_u^1, \ldots, d_u^K) * S_{1:K}  * V_{1:K}^\top * \text{vec}(d_x^1, \ldots, d_x^K)$
>
> with $d_u^j := h(f_w(u_j)) - h( f_{w_*}(u_j))$ and $d_x^i := h(f_w^i) - h(f_{w_*}^i)$ are the discrepancies in prediction at $u_i$ and $x_i$ respectively.
>
> Clearly, with this construction, we only need $K<=N$ inputs to achieve the same result. The inputs are not from the training data and generally lie outside it. Similar constructions can be applied to arbitrary inputs. An “inexact” version where the memory size $M<K$ can also be derived with a bound on the error, strictly decreasing as we increase M to K. We will add this to the next version of the paper.
>
> Q4: Also, the suggested way to select memorized samples is cumbersome
>
> A: No, it is not. It simply requires a forward pass through the model (e.g., for binary classification, we compute $\sigma(f)(1-\sigma(f))$ at function outputs f at multiple locations). We will make this clear in line 169.
>
> Q5: There are existing approaches proposed for sample selection in continual learning as well, e.g. [1],[2]. The authors didn't discuss or compare with any such related work.
>
> A: It is incorrect to say that we didn’t discuss to *any such* related works. See line 247 where we have a discussion with methods such as FROMP [37] and GEM [30]. We have even included a statement in line 262 that our method does not contradict with these methods, rather complements some of them. For continual learning K-priors perform similarly to FROMP, with FROMP performing slightly better for smaller M because it uses the Kernel matrix (line 250) which is essentially the matrix B we discuss in line 176. Note that all these works do not apply to other adaptation tasks such as Change Regularizer or Model-Class. We will add references to the two papers the reviewer mentions. These papers propose computationally-expensive methods for choosing examples to store in memory. In particular, [2] aims to maximize the diversity of the samples in the replay buffer, which is related to choosing examples with high leverage (see lines 162-166).
>
> Q6: And how far this approximation can be to the previous gradient is not analyzed either.
>
> A: This is also wrong. Eq 13 exactly quantifies *how far* the approximation is.
>
> Q7: So, how to achieve the claim of 'recover the exact retrained model to an arbitrary accuracy by choosing a sufficiently large memory of the past data'? For a specified accuracy, how large the memory should be?
>
> A: The result shows that we get arbitrarily close to the batch solution as the memory is increased. The opposite question (how much memory to get a specified accuracy) highly depends on the problem and is the fundamental difficulty of the adaptation. Almost all memory-based methods face this dilemma to which there is no one answer. We do provide an approximate solution based on the GGN matrix approximation.

---

> > ### Comment · Reviewer_5dEe · 2021-08-23
> > **Feedback to authors' response**
> >
> > Q1 is based on Q2, and I still don't feel it is surprising because the optimized weights are known,  the true labels not required is not very attractive.
> >
> > The authors' response to Q3 is interesting and should be added into the paper. Although  representative inputs may be not exactly as the same as raw inputs in the training set, it doesn't mean they are not from the same distribution. On the contrary, they may be closer to the modes of the distribution. So, I still think the statement is problematic and should be rephrased.  Also, could authors explain how do you get  $d_x^1,..., d_x^K$ in the definition of $\beta$ since $K \le N$?
> >
> > Q6, Eq.13 can not answer the question in practice as it requires the data not in the memory. After learning more tasks, can you still keep a track of the approximation error?
> >
> > Q7, So the claimed contribution is not exactly solved.
> >
> > Although I disagree most of the response,  I will increase my score to 4 regarding other reviewers points.

---

> > > ### Author Response · Authors · 2021-08-26
> > > **Response to Reviewer 5dEe**
> > >
> > > Thanks for the additional response. Below are our detailed comments.
> > >
> > > “Q1 is based on Q2, and I still don't feel it is surprising because the optimized weights are known, the true labels not required is not very attractive.”
> > >
> > > A: Your Q2 says “The claims are not supported by the definition of the method” which is incorrect. Our claim is supported, but you seem to not find it surprising. Despite this, the result is still new, because no previous work (including those on knowledge distillation) have shown that such a prior can reconstruct the gradient.
> > >
> > > “The authors' response to Q3 is interesting and should be added into the paper…”
> > >
> > > A: Thanks, we will add this to the paper (as we previously said).
> > >
> > > “Although representative inputs may be not exactly as the same as raw inputs in the training set, it doesn't mean they are not from the same distribution. On the contrary, they may be closer to the modes of the distribution. So, I still think the statement is problematic and should be rephrased”
> > >
> > > A: The reviewer appears to have misread the sentence in line 133 which says “any input locations, even those outside the training data”. The reviewer is is talking about the “data distribution”, which we do not mention. We therefore believe this criticism is wrong and there is no need for rephrasing.
> > >
> > > “Also, could authors explain how do you get ... since $K\leq N$?”
> > >
> > > A: The question is unclear but we will try to answer. $K$ is the rank of $\Phi$, and the theorem shows that we only need to compare predictions at $K\leq N$ (representative) inputs. This is the main point of the theorem, which establishes that the points can lie outside the training data (this is different to the “training data distribution”) and that we can reconstruct the gradient *perfectly* with only $K\leq N$ such inputs.
> > >
> > > “Q6, Eq.13 can not answer the question in practice as it requires the data not in the memory.”
> > >
> > > A: Your Q6 is saying that the error in the approximation is not “analyzed”. The question does not mention anything about “in practice” there. We answered that the error is analyzed in that question.
> > >
> > > “After learning more tasks, can you still keep a track of the approximation error? Q7, So the claimed contribution is not exactly solved”
> > >
> > > A: We do not claim that we can keep track of the approximation error. Rather we only claim that the error goes to zero as the memory size is increased. Please see the line in the abstract where we say we “can often recover the exact retrained model to an arbitrary accuracy by choosing a sufficiently large memory of the past data”.

---

### Official Review · Reviewer_2Zuk · 2021-07-17

**Rating:** 7
**Confidence:** 3

**Summary:**

- This paper tackles an important problem: Suppose we have trained a model on a dataset, but we wish to adapt it quickly. For example, we might add new training points, delete training points, or wish to change the model and regularization. We could re-train a model, but that's slow - can we get a similar result but quicker and using less memory?
- This paper proposes K-prior: regularizing the model towards the original model in both weight space and predictions.
- They show that training on the K-prior objective on the entire dataset recovers the original model - which would not be true if we only have weight space regularization, or only have prediction regularization.
- In practice, to save memory, they only form the prior on a few points, and they describe heuristics to choose these points, with some mathematical justification
- They show that this methods work well in practice, compared to just replaying the points


**Limitations And Societal Impact:**

Seems fine to me!

**Main Review:**

After rebuttal: added comments below

---------------------------------------------------------------------------------------
Strengths
- The paper is well written. It seems well executed, technically sound, and reports results honestly and thoroughly.
- The idea of K-priors is nice - it's pretty interesting that if you keep both the weight space regularization and prediction space regularization, then you can recover the original model. The math is simple, but I tend to view that as a strength
- Experimental results on UCI adult, MNIST, USPS, and CIFAR-10, show promise. It's interesting that the K-prior method does better than replaying the loss on the same set of points they use for the K-prior regularization. They also include ablations on using only the weight regularization.

Questions and suggestions for improvement:
- It would be more compelling if they include baselines from other papers. They say that prior methods focus on specific tasks, like adding points, or removing points, and don't do all the adaptations they do. However, it would still be nice to see how K-priors compare with methods specialized for adding new points. I'm not too familiar with the area or relevant baselines though.
- The paper says that the main difference between "Replay and K-priors is that the former uses the true label while K-priors use model-predictions." - does replay also add weight regularization besides replaying the loss?
- Just to confirm, what was the tuning protocol to choose tau, the weighting between the weight and prediction regularization?
- I think it would be good to discuss in more detail why K-priors do better than just replaying the loss. I believe replaying the loss would satisfy similar guarantees as K-priors? For example, in equation 10 and Theorem 1, if we replaced K(w) by sum_{i \in D} l_i(w) + R(w). So what's the benefit of K-priors?
- It seems like one benefit of K-priors could be that we get more (bits of) information from the model prediction than from the label, similar to in distillation, since the model prediction is a probability vector but the label is just one of C values, where C is the number of classes. The paper could be stronger if it explored these ideas and compared K-priors with replays.
- Their method of selecting points, for linear models, seems to roughly correspond to choosing points close to the margin. Is that correct?
- The experiments could be stronger if they did some grid search of hyperparameters for each method. For example, maybe weight priors need a stronger weight because they don't have the prediction space regularization? So ideally if they do the same grid search of hyperparameters for each method, and then compare them, it would be more convincing.

nit: In line 301, a neural network with 1 hidden layer is not deep, so remove deep :)

Overall, I think it's a nice idea, that's well motivated, and fairly well executed, so would be good to have in the conference!


**Time Spent Reviewing:**

4 hours

---

> ### Author Response · Authors · 2021-08-10
> **Response to Reviewer 2Zuk**
>
> Q1: It would be more compelling if they include baselines from other papers.
>
> A: In Sec 4, we provide theoretical results showing equivalence and connections to many such baselines (e.g., SVMs, GPs, KD, and continual-learning methods). Empirical results simply confirm these connections, which we have verified for some baselines (e.g., weight-priors, KD, and CL methods). We did not add all of them due to space constraints, but can include CL and GP methods if the reviewer insists.
>
> Q2: Does replay also add weight regularization besides replaying the loss?
>
> A: Replay employs the standard L2 weight regularizer (penalizes the norm of $w$). This is different to the K-priors weight “divergence” term in Eq. 7 (which penalizes the norm of $w-w_*$).
>
> Q3: Just to confirm, what was the tuning protocol to choose $\tau$?
>
> A: We use $\tau=1$ for all experiments. We also provide an ablation in Appendix D.1 (Figure 4), where we try $\tau=N/M$, where $N$ is the past data size, and $M$ is the number of datapoints stored in memory.
>
> Q4: I think it would be good to discuss in more detail why K-priors do better than just replaying the loss….. It seems like one benefit of K-priors could be that we get more (bits of) information from the model prediction than from the label, similar to in distillation, since the model prediction is a probability vector but the label is just one of C values, where C is the number of classes. The paper could be stronger if it explored these ideas and compared K-priors with replays.
>
> A: You are exactly right! When using all past data, K-priors can reconstruct the exact gradient, but for smaller memory size, they provide more information than Replay with the true labels. This is why they give much better results (see line 274). This is the first main point of the paper (see the para at line 103). We will further clarify the writing to better emphasize this point.
>
> Q5: Their method of selecting points, for linear models, seems to roughly correspond to choosing points close to the margin. Is that correct?
>
> A: Yes, this is roughly the idea. The first equality in Eq. 13 suggests picking points where the predictions disagree the most. The points close to the margin are the most likely to be such candidates. This is very similar to the role of support vectors in incremental SVM. This is mentioned in line 229.
>
> Q6: if they did some grid search of hyperparameters for each method. For example, maybe weight priors need a stronger weight
>
> A: We could add a $\tau$ parameter for weight-priors, and this might improve performance slightly. However, such scalar weighting will not fix weight-priors in general. This is clearly shown in Figure 3(a) and explained in lines 306-310. The main reason is that weight-priors use a “stale” $h’(f^i_{w_*})$ (evaluated at the “old” $w_*$), which is difficult to fix through $\tau$.
>
> Q7:​​ “with 1 hidden layer is not deep, so remove deep :)”
>
> A: We will change this.

---

> > ### Comment · Reviewer_2Zuk · 2021-09-01
> > **Thanks for the response**
> >
> > Thanks for the response, this all sounds good, and I stand by my accept recommendation. I think adding a couple more baselines could be good to connect it more to the continual learning literature, although it's not critical.

---

### Author Response · Authors · 2021-08-10
**General comments for AC and the reviewers**

We thank the reviewers for their feedback.

Reviewer 5dEe has given a very low score of 3 to the paper, but their arguments undermine the importance of our results on the “gradient reconstruction”. We strongly disagree with this, and the comments by Reviewer 2Zuk and wqo5 do so too. We have clarified mistakes or misunderstanding in many of their comments. We have also provided an additional result how inputs *outside* the training data can be used to reconstruct the gradients using SVD (see Q3 in the response https://openreview.net/forum?id=_cXX-Dr7sf0&noteId=0Btcb97ce_-). We request the reviewer to reconsider their score.

Reviewer AAdt has given a score of 5. The reasons behind this score are not completely clear to us. It is likely that there are a few misunderstandings, and we hope that our clarification in Q7 will help the reviewer reconsider their score (https://openreview.net/forum?id=_cXX-Dr7sf0&noteId=h5K7gS0npL1). We are happy to engage in further discussions to clarify any other questions they might have.

The reviewers generally appreciated connections to Knowledge Distillation. Since submission, we have improved our results shown in Fig. 3(d). These improvements are summarized in Q3 for Reviewer wqo5 (https://openreview.net/forum?id=_cXX-Dr7sf0&noteId=GbaQmPnN0M).

Finally, we find that reviews generally have not commented on our contribution regarding the “recovery and generalization of many existing, but seemingly-unrelated, adaptation strategies”, specifically connections to SVMs, Bayesian learning, and GPs. Comments about this part are entirely absent from the review of Reviewers 5dEe and AAdt. We hope that reviewers will see the importance of this contribution and reconsider their scores.

---

### Decision · Program_Chairs · 2021-09-27

**Decision:**

Accept (Poster)

**Comment:**

The paper presents a family of approaches for adapting pre-trained models to a variety of changes to the model architecture, training data or other aspects of the training setup. Given the cost of training models from scratch and the generality of the presented approach, this is a highly relevant piece of work with potential for impact.

After the discussion, three reviews recommend acceptance and one review recommends rejection. The negative review points to certain issues with the current version, but doesn't convince me that these issues rise to the level of rejection. After considering both positive and negative points, I lean towards recommending acceptance.

**Comments to the authors**: After reading the paper, I was left with the impression that clarity could be improved, and that there are places where details are missing. I see that some of these missing details and some additional experiments were supplied during the discussion with the reviewers. I would like to see these, together with any other reviewer feedback, incorporated into the final version of the paper.